# Predicting LLM Hallucination Risk from Entity Frequency: A Rate-Distortion Perspective

## Abstract

Large language models struggle with factual hallucinations, and mitigating them typically requires executing the model to assess uncertainty. We introduce a query-dependent rate–distortion framework showing that factual accuracy follows a predictable, sigmoidal "knowledge cliff" governed by exposure-linked entity prominence. Below a critical frequency $f_{\text{crit}}$, the model rarely recalls facts reliably; above this threshold, accuracy rises sharply. We present five core findings based on this framework. First, mapping this frequency response yields an accurate, pre-inference risk score (no additional model execution); we formally prove that this pre-inference metric achieves $> 99\%$ of the theoretical upper bound for any frequency-only predictor. Second, this cliff is heterogeneous, with $f_{\text{crit}}$ varying by more than an order of magnitude ($27$–$76\times$) across relation types. Integrating these structural metadata creates a classifier that outperforms the LLM's own post-inference confidence scores in the rare-entity tail. Third, we establish a sample-complexity bound demonstrating that the calibration data required to locate this cliff under Zipfian sampling is independent of its location. Fourth, we isolate the effect of model scale using the Qwen2.5-Instruct family ($0.5\text{B}$–$14\text{B}$, fixed corpus), revealing a power-law relationship ($f_{\text{crit}} \propto P^{-\beta}$ with $\hat{\beta} \approx 0.5$) where a $10\times$ parameter increase yields reliable recall for entities only $\sim 3\times$ rarer. Finally, we show that these metadata-driven signals establish a favorable budget–utility frontier for retrieval routing. These results suggest that much of a model's factual reliability can be anticipated from query properties before generation, offering a basis for pre-inference triage in retrieval-augmented systems.

## 1 Introduction

Large language models (LLMs) produce fluent, confident text regardless of whether they possess the underlying factual knowledge. This creates a fundamental reliability problem: for any given query, it is unclear *a priori* whether the model will produce a correct answer or a hallucination. The prevailing approach to hallucination detection operates post-hoc such as analyzing output logits, confidence scores, or semantic consistency *after* generation (Kadavath et al., 2022; Manakul et al., 2023; Kuhn et al., 2023). This is expensive at scale since for a system handling millions of queries per day, detecting hallucination after generation wastes both computation and user trust. In this work, we demonstrate that for factual question-answering about entities, a key determinant of hallucination is not the model's generation process but a property of the query itself: how frequently the subject entity appeared in pretraining data. This observation has been noted empirically (Kandpal et al., 2023; Mallen et al., 2023; Sun et al., 2024), but lacks a theoretical framework that explains the *shape* of the failure mode, predicts which parameters are invariant across models, and yields a practical detection tool.

**Contributions.** We develop a query-dependent rate–distortion framework for LLM factual reliability and derive five empirically validated results. An LLM's chance of recalling a fact correctly depends sharply on how often the entity appeared in training; the threshold for "often enough" varies with the type of question; larger models lower the threshold but never eliminate the long tail; and all of this can be predicted before running the model. In detail:

1. **A predictable accuracy boundary from limited capacity.** Because a model has finite room to store facts, it reliably learns common entities and largely fails on rare ones, with a well-defined cutoff between. Modeling factual recall as lossy compression under a finite budget over a Zipfian entity distribution, we derive a sigmoidal phase transition (the "knowledge cliff") in accuracy versus exposure, characterized by a critical frequency $f_{\text{crit}}$ (Section 3).

2. **A pre-inference risk score with a provable ceiling.** We can score a query's hallucination risk *before* running the model, using only entity frequency (AUROC = 0.810), and we derive a closed-form ceiling on the discrimination any frequency-only score can achieve, increasing in cliff steepness (0.816 for Mistral-7B, 99% attained). Adding model confidence is complementary and surpasses the frequency-only ceiling (Sections 4 and 6.4).

3. **Question type sharpens individual predictions.** The cutoff differs by question type, and relation structure helps most where frequency alone cannot, in the rare-entity tail. The cliff location varies by more than an order of magnitude across relations (27–76×), and relation type dominates discrimination in the tail. The best fully pre-inference predictor reaches AUROC = 0.842 and yields competitive retrieval-routing frontiers (Sections 4 and 6.6).

4. **Bigger models help, but cannot fix the tail.** Scaling lowers the cutoff so rarer entities become reliable, but gradually, and the extreme tail stays unreliable at every size tested. On the Qwen2.5-Instruct family (0.5B–14B, fixed corpus) we find $f_{\text{crit}} \propto P^{-\beta}$ with $\hat{\beta} \approx 0.5$ ($R_w^2 = 0.97$): a 10× larger model recalls entities only $\sim 3\times$ rarer (Section 6.7).

5. **Calibrating the predictor is cheap.** Locating the cutoff for a new model or domain takes few labeled examples, and steeper cutoffs are easier to locate. We derive the Fisher information for $x_{\text{crit}} = \log f_{\text{crit}}$ and a closed-form sample-complexity bound: under Zipfian sampling the required $n$ is independent of $f_{\text{crit}}$ and scales inversely with the cliff's steepness, matching the empirical ECE $< 0.10$ at $\sim 75$ samples (Section 5).

## 2 Related Work

That LLMs struggle with rare entities is well established. Neural scaling laws (Kaplan et al., 2020; Hoffmann et al., 2022) predict power-law loss decrease in model size and data, but these aggregate trends mask entity-level heterogeneity. Two lines of work are closest to ours. Kandpal et al. (2023) establish that factual accuracy correlates (causally as well as correlationally) with the number of relevant pretraining documents for a fact, and that retrieval augmentation reduces this dependence. Mallen et al. (2023) introduce PopQA, use Wikipedia popularity (including page views) as a long-tailness proxy, and show retrieval helps low-popularity entities most; the Head-to-Tail benchmark (Sun et al., 2024) documents the same head-to-tail collapse at larger scale. Our work is complementary to these empirical findings. Rather than treating the frequency–accuracy relationship as an empirical regularity, we propose a *query-dependent rate–distortion* account that predicts *why* the transition is sharp (threshold distortion), *where* it occurs (a critical exposure $f_{\text{crit}}$), and, how the boundary transfers and scales across models. In other words, prior work establishes the long-tail phenomenon and the utility of exposure proxies; we supply an RD account that explains it and derive decision-relevant limits and policies from it.

A natural response to unreliable outputs is to detect hallucinations after generation. Kadavath et al. (2022) use token-level self-evaluation (P(True)); Kuhn et al. (2023) propose semantic entropy (consistency across sampled outputs), extended by Farquhar et al. (2024) to confabulation detection; Manakul et al. (2023) introduce SelfCheckGPT via self-consistency; and Min et al. (2023) develop FActScore for long-form factuality. Probe-based approaches train classifiers on internal representations (Azaria & Mitchell, 2023; Li et al., 2024), and Mahaut et al. (2024) systematize factual-confidence estimation across benchmarks including PopQA. All of these require running the model first, so detection cost scales with query volume and the hallucination is generated (and possibly served) before it is caught. Our risk score instead depends only on query-side properties, exposure and relation structure, and is computed *pre-inference* with no additional model execution.

This pre-inference risk score naturally connects to the question of *when* to augment generation with retrieval. Adaptive retrieval methods such as Self-RAG (Asai et al., 2024) and FLARE (Jiang et al., 2023b) decide

during generation whether to retrieve, but still require partial inference to make the routing decision. Similarly, RouteLLM (Ong et al., 2025) learns a routing function to dispatch queries between systems, but requires learned routing machinery. Our framework provides a routing signal available before any inference begins, and we show that optimal budgeted routing under a hard retrieval cap has a simple closed-form solution: retrieve for the queries with the largest expected gain.

Rate–distortion theory (Cover & Thomas, 2006) has been applied to learning and compression in neural networks: Shwartz-Ziv & Tishby (2017) use the information bottleneck to analyze deep learning, Nokleby et al. (2016) derive rate–distortion bounds on achievable Bayes risk in supervised learning, and Delétang et al. (2024) formalize language modeling as compression, a natural foundation for our formulation. In the memorization literature, Carlini et al. (2023) establish log-linear relationships between memorization and both capacity and duplication count; Tirumala et al. (2022) observe sigmoidal memorization curves over training; and Allen-Zhu & Li (2025) estimate ∼2 bits of extractable knowledge per parameter with heterogeneous extractability across knowledge types. These motivate our three ingredients: logarithmic rate in exposure, heterogeneous storage difficulty, and a sharp transition under threshold distortion. From a representational angle, Merullo et al. (2025) find that linear representations of subject–relation–object facts form only above a co-occurrence-frequency threshold (∼1–2K in 7B models, higher in smaller ones) that varies by relation type, an internal-mechanism counterpart to the cliff, its capacity scaling, and its relation heterogeneity that we characterize behaviorally.

Most directly related to our theoretical contribution is Kalai & Vempala (2024), who proved that calibrated language models *must* hallucinate on singleton facts at a rate tied to Good–Turing estimation, and the follow-up (Kalai et al., 2025), which argues that standard benchmarks score a wrong answer and an abstention ("I don't know") identically, giving models an incentive to guess confidently rather than admit uncertainty. The binary scoring they identify is exactly the threshold distortion of our Assumption 1, $d(x, \hat{x}) = \mathbf{1}[\hat{x} \neq x]$, and it is this threshold structure, not finite capacity alone, that produces the sharp transition: replacing it with a smooth measure (e.g. log-loss) would soften the cliff into a gradual rolloff (Section 7). The two views are complementary: they explain why models guess, and we explain why the guessing has a cliff.

Concurrently, Guo & Li (2026) also apply rate–distortion theory to LLM hallucination, formalizing it as membership testing: in a sparse fact universe, the space-optimal strategy for a capacity-constrained model is to assign high confidence to non-facts rather than abstain. Where their analysis treats a binary fact/non-fact setting, ours extends the rate–distortion lens to the continuous spectrum of entity exposure with its scaling laws across model size, and turns the theoretical limits into a deployable pre-inference routing metric.

## 3 Theoretical Framework

### 3.1 Preliminaries and Notation

Before developing the framework, we define the concepts used throughout.

Entity frequencies in natural corpora are highly skewed. We model this with the *Zipf distribution*, $f_k \propto k^{-\alpha}$, where $k$ is an entity's frequency rank and $\alpha > 0$ controls skew ($\alpha \approx 1$ empirically; Zipf, 1949; Piantadosi, 2014). Fact frequencies in LLM pretraining corpora are correspondingly heavy-tailed, and this structure governs factual recall (Kandpal et al., 2023; Merullo et al., 2025). We treat factual storage as lossy compression over this distribution. The *rate* $r_k$ is the bits of capacity allocated to entity $k$, and the *distortion* $d_k(r_k) \in [0, 1]$ is the probability the model fails to recall $k$'s fact given $r_k$ bits, decreasing in $r_k$. When a single distortion function is shared across entities (the homogeneous case) we write it $D(r)$, and take it to be convex and decreasing, encoding diminishing returns: the first bits allocated to a fact reduce its error the most, and each additional bit helps less. The limiting step function (a fact is stored, $d = 0$, or not, $d = 1$) is *threshold distortion*. The capacity-allocation problem (Definition 1) minimizes $\sum_k q_k d_k(r_k)$ subject to $\sum_k r_k \leq R_{\text{total}}$.

Our central object is the *knowledge cliff*: a sigmoidal rise in accuracy as a function of log-frequency, with inflection at a *critical frequency* $f_{\text{crit}}$ and steepness governed by $\kappa$. We refer to $f \ll f_{\text{crit}}$ as the low-frequency regime, in which the entity received effectively zero allocated rate, so the model cannot recall its fact and reverts to a generic prior, and to $f \gg f_{\text{crit}}$ as the high-frequency regime, in which the fact is reliably encoded.

Finally, we use two standard evaluation notions, which are distinct. A risk score is *calibrated* if its predicted probabilities match empirical frequencies (e.g., queries assigned risk 0.3 are wrong 30% of the time), and we measure miscalibration by the *Expected Calibration Error* (ECE), the average gap between predicted and observed error rates across probability bins. Separately, a score *discriminates* if it ranks hallucinations above correct answers, measured by the *Area Under the ROC Curve* (AUROC): the probability that a randomly chosen hallucination receives a higher risk score than a randomly chosen correct answer. The two are independent in that a monotone rescaling can fix calibration without changing AUROC (Remark 3).

## 3.2 Setup: Factual Recall as Lossy Compression

We adapt the rate–distortion framework to factual recall rather than applying it literally: "distortion" is the probability of hallucination, and the distortion–rate curve is posited (and validated in Section 6) rather than derived from a source code. An LLM $\mathcal{M}$ is trained on the entity set $\mathcal{K} = \{1, \dots, K\}$, where each query $x_k$ carries a relation label $R_k \in \mathcal{R}$ (e.g. `capital-of`, `born-in`) whose correct answer instantiates that predicate.

Training must encode these facts into a fixed parameter budget. With $\sim 2$ bits of extractable knowledge per parameter (Allen-Zhu & Li, 2025), a 7B model has effective capacity $R_{\text{total}} \approx 14 \times 10^9$ bits, far below the lossless requirement for millions of entity–fact pairs. Training therefore performs implicit *lossy compression*: frequent facts are encoded redundantly while rare facts receive fragile representations, so the parameters act as a finite-capacity channel between corpus and queries.

**Definition 1** (Query-dependent rate-distortion). *With entity distribution $q_k \propto f_k$ (the distribution SGD implicitly optimizes) and per-entity distortion $d_k(r_k)$, the capacity-allocation problem is*

$$\min_{\{r_k\}} \sum_{k=1}^{K} q_k \, d_k(r_k) \quad s.t. \quad \sum_{k=1}^{K} r_k \leq R_{\text{total}}, \qquad R_{\text{total}} \approx 2P, \tag{1}$$

*for a model with $P$ parameters (Allen-Zhu & Li, 2025). The formulation is query-dependent: each entity has its own distortion $d_k$, weighted by exposure $q_k$.*

SGD does not literally solve this water-filling problem but approximates it: high-frequency entities are encoded redundantly and recalled robustly, low-frequency entities get fragile representations. The key question is the *shape* of the resulting transition. The setup makes three idealizations. The Zipfian frequency form is inherited from the corpus, not assumed, and is among the most robust regularities in language statistics (Zipf, 1949; Piantadosi, 2014); we verify insensitivity to its precise form via the Mandelbrot–Zipf generalization (Remark 2). The finite set $\mathcal{K}$ indexes the Zipf-ordered entities and makes no claim about the true number of world facts, and the capacity figure serves only to establish that $R_{\text{total}}$ is finite and sub-lossless.

**Proposition 2** (Knowledge Cliff via Convex Rate–Distortion). *Let $D : [0, \infty) \to [0, 1]$ be differentiable, strictly convex, and strictly decreasing, and let entity frequencies $f_k$ be Zipfian with weights $q_k = f_k / \sum_j f_j$. In the homogeneous allocation problem with budget $R_{\text{total}}$, the optimal hallucination probability exhibits a sharp cutoff:*

$$P(\text{error} \mid f) = \begin{cases} D(0) & \text{if } f \leq f_{\text{crit}} \\ D\left((D')^{-1}\left(-\frac{\tilde{\lambda}}{f}\right)\right) & \text{if } f > f_{\text{crit}} \end{cases} \tag{2}$$

*where $f_{\text{crit}} = -\tilde{\lambda}/D'(0)$ and $\tilde{\lambda} > 0$ is determined by the budget $R_{\text{total}}$.*

*Proof.* Take the homogeneous case of Definition 1, $d_k(\cdot) \equiv D(\cdot)$, giving the allocation problem $\min_{\{r_k \geq 0\}} \sum_k q_k D(r_k)$ s.t. $\sum_k r_k \leq R_{\text{total}}$. The KKT conditions give $q_k D'(r_k) + \lambda - \mu_k = 0$ with $\mu_k \geq 0$, $\mu_k r_k = 0$. At an interior optimum ($r_k^* > 0$, so $\mu_k = 0$), $D'(r_k^*) = -\lambda/q_k$; since $D$ is strictly convex, $D'$ is strictly increasing and invertible, so $r_k^* = (D')^{-1}(-\lambda/q_k)$. The non-negativity constraint binds ($r_k^* = 0$) exactly when $q_k \leq -\lambda/D'(0)$. Writing $q_k = f_k / \sum_j f_j$ and $\tilde{\lambda} := \lambda \sum_j f_j > 0$ gives the cutoff $f_{\text{crit}} := -\tilde{\lambda}/D'(0)$. Entities below it receive zero capacity and suffer distortion $D(0)$; entities above it get $r_k^* = (D')^{-1}(-\tilde{\lambda}/f_k)$, yielding $P(\text{error} \mid f) = D((D')^{-1}(-\tilde{\lambda}/f))$, with $\tilde{\lambda}$ fixed by $\sum_{k: f_k > f_{\text{crit}}} (D')^{-1}(-\tilde{\lambda}/f_k) = R_{\text{total}}$. $\qquad\square$

Proposition 2 assumes a smooth, strictly convex $D(R)$. In the next section, we derive the sigmoid from a complementary starting point: threshold distortion (Assumption 1), which is a step function and therefore *not* strictly convex. The two results are reconciled by noting that heterogeneous storage difficulty (Assumption 2) averages over a population of threshold distortion functions, producing an *effective* population-level $D(R)$ that is smooth and convex. The sigmoid of proposition 4 can thus be viewed as the special case of proposition 2 where the effective distortion-rate curve arises from a logistic mixture of step functions.

### 3.3 The Knowledge Cliff as Phase Transition

The shape of the distortion function $d_k(r_k)$ determines whether the transition from reliable to unreliable recall is gradual or sharp. We make three modeling assumptions, each grounded in independent empirical findings from the memorization literature.

**Assumption 1 (Threshold distortion).** Storing a fact requires a minimum rate: $d_k(r_k) = 0$ if $r_k \geq r_{\min,k}$ and $d_k(r_k) = 1$ otherwise. Factual recall is approximately all-or-nothing, so a partial representation is of little value.

**Assumption 2 (Heterogeneous storage difficulty).** Different facts require different minimum rates, $r_{\min,k} = r_0 + \eta_k/\kappa$, where $\eta_k$ captures how much harder or easier fact $k$ is to store than average, independent of its frequency. We take $\eta_k$ to follow a standard logistic distribution, with $1/\kappa$ setting the spread; Allen-Zhu & Li (2025) find that different knowledge types require differing amounts of capacity to become extractable.

**Assumption 3 (Logarithmic rate allocation).** Training on $f_k$ occurrences of entity $k$ provides an effective rate $r_k^{\mathrm{eff}} \propto \log f_k$. This follows the log-linear relationship between memorization and duplication count established by Carlini et al. (2023).

### 3.4 Deriving Logarithmic Rate Allocation from Gradient Dynamics

Assumptions 1 and 2 are empirically grounded (Allen-Zhu & Li, 2025), and the heterogeneity of Assumption 2 is borne out by the order-of-magnitude spread in relation-specific $f_{\mathrm{crit}}$ (Section 6.6). Assumption 3 rests on the empirical log-linear relationship observed by Carlini et al. (2023), and is directly supported on our data: accuracy fits substantially better against $\log f$ than raw $f$ ($R^2 = 0.958$ vs. $0.817$). We now derive this logarithmic encoding from gradient dynamics.

**Proposition 3** (Gradient saturation implies logarithmic encoding). *Fix an entity $k$ that appears in $M$ supervised training steps, and let $u^{(m)} := f_k(\theta^{(m)})$ be the pre-softmax margin for the correct answer after step $m$ (the logit of the correct token minus that of the strongest competitor). Suppose updates use the logistic loss $\mathcal{L}(u) = -\log \sigma(u)$ by gradient descent, $\theta^{(m+1)} = \theta^{(m)} - \eta \nabla_\theta \mathcal{L}(u^{(m)})$, with squared gradient norms bounded in $[g_{\min}, g_{\max}]$, $0 < g_{\min} \leq g_{\max} < \infty$. If $\eta M \to \infty$ and $M\eta^2 \to 0$ (e.g. $\eta = M^{-3/4}$), then*

$$u^{(M)} = \log(\eta M) + O(1) + O(M\eta^2). \tag{3}$$

*If the entity is sampled at rate $q_k \propto f_k$ over $T$ steps, its occurrence count $M$ is random with mean $\mathbb{E}[M] = q_k T$; by concentration, $\log M = \log(q_k T) + o(1)$ with high probability, so $u^{(M)} = \log f_k + \text{const}$, establishing logarithmic rate allocation.*

*Proof.* See Section A. $\square$

As an explicit modeling assumption, we identify the accumulated margin $u^{(M)}$ with the effective allocated rate $r_k^{\mathrm{eff}}$: the margin is the log-odds the model assigns to the correct answer, which equals the code-length reduction (in nats) over a uniform prior, the natural per-fact analogue of rate. The downstream theory uses only that $r_k^{\mathrm{eff}}$ is monotone in $u^{(M)}$, not the exact identification.

**Proposition 4** (Sigmoid phase transition). *Under Assumptions 1–3, expected accuracy is an affine sigmoid in log-frequency:*

$$\mathrm{acc}(f) = a_{\min} + (a_{\max} - a_{\min}) \cdot \sigma\big(\kappa(\log f - \log f_{\mathrm{crit}})\big), \tag{4}$$

*where $f_{\mathrm{crit}}$ is the inflection point, $\kappa$ the inverse scale of storage heterogeneity (Assumption 2), and $a_{\min} \leq a_{\max}$ in $[0,1]$ the asymptotic accuracies as $f \to 0$ and $f \to \infty$. All four are fitted, not fixed in advance.*

*Proof.* See Section A. □

*Note.* Only the specific logistic form of (4) uses Assumption 2's logistic choice; the cliff itself is distribution-free. For any continuous heterogeneity distribution with CDF $F$, the same argument gives $\mathrm{acc}(f) = a_{\min} + (a_{\max} - a_{\min}) F(\kappa(\log f - \log f_{\mathrm{crit}}))$, a smooth monotone transition centered at $f_{\mathrm{crit}}$. The logistic choice selects $F = \sigma$, yielding the closed form we fit.

**Remark 1** (Choice of distribution and steepness). *The logistic form of the storage heterogeneity (Assumption 2) is chosen for convenience, and our conclusions are robust to it. The candidate forms differ mainly in their tails, where the data is sparse, so refitting with six alternatives (Gaussian, Laplace, Gumbel, Cauchy, a Gaussian mixture, and logistic) leaves the fits statistically indistinguishable (maximum $\Delta\mathrm{AIC} = 0.66$; Burnham & Anderson 2002) and the cliff location stable (*$\log f_{\mathrm{crit}}$ *spans* 0.11 *decades, where one decade is a factor of* 10 *in frequency, i.e. one unit of* $\log_{10} f$*). The raw* $\kappa$ *varies across forms only because their CDFs have different slopes at the origin; the well-determined, distribution-free quantity is the central steepness* $s^* = (a_{\max} - a_{\min})\kappa f(0)$*, which is stable (*$s^* \in [0.90, 1.10]$*, CV = 7.9%). Empirically* $\kappa$ *has* CV > 1 *across random splits, suggesting storage heterogeneity is not captured by a single scale and that relation type adds a second axis of variation (Section 6.6). Full details are in Section B.*

We note that the critical frequency $f_{\mathrm{crit}}$ we discussed so far emerges from the interaction between the capacity constraint and the Zipfian distribution: entities with $f_k \gg f_{\mathrm{crit}}$ have been seen enough times to be reliably stored, while entities with $f_k \ll f_{\mathrm{crit}}$ fall below the coverage threshold. We next characterize this as a function of total capacity and training corpus.

**Proposition 5** (Data-determined cliff). *Under the threshold allocation of Proposition 4, the model stores the* $K^* = \lfloor R_{\mathrm{total}}/r_{\min} \rfloor$ *most frequent entities, and the critical frequency is the frequency of the marginal stored entity,* $f_{\mathrm{crit}} = f_{K^*}$*:*

$$f_{\mathrm{crit}} = \frac{N}{H_K^{(\alpha)}}\left(\frac{R_{\mathrm{total}}}{r_{\min}}\right)^{-\alpha}, \qquad \log f_{\mathrm{crit}} = \log N - \log H_K^{(\alpha)} - \alpha \log R_{\mathrm{total}} + \alpha \log r_{\min}, \qquad (5)$$

*where* $N$ *is the total training tokens and* $H_K^{(\alpha)} = \sum_{j=1}^{K} j^{-\alpha}$*.*

Thus $f_{\mathrm{crit}}$ is determined by the corpus $(N, K, \alpha)$ and capacity ($R_{\mathrm{total}}$), not by architecture beyond its effect on storage capacity; the $-\alpha \log R_{\mathrm{total}}$ term governs the log–log scaling observed in Section 6.7.

*Proof.* See Section A. □

**Remark 2** (Extension to Mandelbrot-Zipf). *The derivation generalizes to the Mandelbrot-Zipf distribution by replacing* $k^{-\alpha}$ *with* $(k + q)^{-\alpha}$*, where* $q \geq 0$ *flattens the distribution at high-frequency ranks. The harmonic number becomes the Hurwitz-type sum* $H_{K,q}^{(\alpha)} = \sum_{j=1}^{K}(j + q)^{-\alpha}$*. Since* $q$ *is constant relative to* $N$ *and* $K$*, the asymptotic behavior of* $\log H_{K,q}^{(\alpha)}$ *remains dominated by* $K$ *(and thus* $N^{\beta}$*), so the scaling relation in Equation* (5) *holds with* $q$ *affecting only the constant offset.*

Proposition 5 yields three predictions. First, models of similar capacity and coverage should share a similar $f_{\mathrm{crit}}$; Falcon-7B and Qwen-2.5-7B cluster at $f_{\mathrm{crit}} \approx 51$–66K, across differing architectures and corpora. Second, $f_{\mathrm{crit}}$ *increases* with raw corpus size $N$ at fixed capacity, so the relevant cross-model quantity is the per-entity exposure $\bar{f} := N/H_K^{(\alpha)}$, not $N$. Third, $f_{\mathrm{crit}} \propto R_{\mathrm{total}}^{-\alpha}$: we obtain $\hat{\alpha} = 0.52 \pm 0.06$ ($R_w^2 = 0.97$) on the Qwen2.5-Instruct family (0.5B–14B, fixed 18T-token corpus), robust to leave-one-model-out and bootstrap perturbations (Section 6.7). We also note that for matched-capacity models the capacity factor cancels in ratios, $f_{\mathrm{crit}}^{(A)}/f_{\mathrm{crit}}^{(B)} = \bar{f}_A/\bar{f}_B$, consistent with our three 7B models: Falcon and Qwen cluster despite a $12\times$ gap in token count (ratio 1.30), plausibly because Qwen's 18T multilingual tokens give English exposure comparable to Falcon's 1.5T English-focused corpus, while Mistral sits $5.2\times$ lower.

**Note on the overconfidence at low frequency.** On rare entities, model confidence does not track the falling accuracy: it remains near its high-frequency level while accuracy drops to the floor $a_{\min}$, so the

expected calibration error grows in the rare-entity tail (Section 6.4). Two ingredients explain this. First, the rate–distortion picture predicts *what* the model outputs: at $f \ll f_{\mathrm{crit}}$ the entity receives effectively zero rate, so the output reverts to the relation prior $\pi_R$ (the zero-rate distribution defined above), independent of the true answer. Second, the binary-scoring incentive identified by Kalai et al. (2025) explains *why that reverted output is confident rather than hedged*: when a wrong answer and an abstention score identically, the reward-maximizing behavior is to commit to a single best guess at full weight rather than spread mass across alternatives. The model therefore generates a confident, fluent, prior-conditioned answer whose certainty reflects the peakedness of $\pi_R$, not correctness. This is the mechanism behind the high-confidence false positives in the rare-entity tail, and it is why the model's own confidence cannot substitute for the frequency signal there.

**Remark 3** (Calibration vs. Discrimination). *Because confidence is elevated across the entire low-frequency regime, global post-hoc methods like temperature scaling cannot solve the hallucination gap. Monotone transformations improve global calibration metrics (ECE) but leave example ordering, and thus discrimination (AUROC), unchanged. Correcting this requires a signal that distinguishes reliably stored facts from unstored ones, which is why our frequency-based predictor dominates post-hoc scaling empirically (Section 6).*

**Remark 4** (Connection to membership testing). *Guo & Li (2026) formulate hallucination as rate-distortion for binary membership testing, where a space-constrained model must assign high confidence to some non-facts. This is the special case of our framework with a flat exposure prior ($f_k = $ const): without a Zipfian ordering, the capacity bottleneck forces zero-bit allocation to a random subset, and the reversion to the relation prior $\pi_R$ (discussed above) is consistent with their high-confidence false positives.*

## 4 Pre-Inference Risk Scoring and Retrieval Routing

The rate–distortion framework in Section 3 implies a frequency-driven transition in factual reliability. In particular, the cliff model yields a pre-inference prediction of error for an entity $k$ with frequency $f_k$:

$$\widehat{d}_{0,k} := \Pr(\mathrm{error} \mid f_k) \approx \sigma\big(\kappa(\log f_{\mathrm{crit}} - \log f_k)\big). \tag{6}$$

Routing turns this distortion proxy into a policy: under limited retrieval capacity, allocate retrieval to the queries where it most reduces expected distortion. We consider three actions, direct generation (noRAG, $i = 0$), retrieval-augmented generation (RAG, $i = 1$), and abstention ($\perp$, $i = 2$, returning "I don't know"), and let $x_k$ be the query for entity $k$ with relation $R_k$. Retrieval carries a relation-conditioned residual risk,

$$\widehat{d}_{1,k} := \widehat{d}_1(R_k), \tag{7}$$

a static pre-inference lookup estimated from logs. Using the same utility convention as our experiments (correct $+1$, incorrect $-\beta$, retrieval cost $c_{\mathrm{RAG}} > 0$, abstention $-\eta$), the expected utilities are $u_{0,k} = 1 - (1 + \beta)\widehat{d}_{0,k}$, $u_{1,k} = 1 - (1 + \beta)\widehat{d}_{1,k} - c_{\mathrm{RAG}}$, and $u_{2,k} = -\eta$. With query mass $q_k \propto f_k$ and a hard cap $b_{\mathrm{RAG}}$ on the retrieval fraction, a deterministic policy $T : \{1, \ldots, K\} \to \{0, 1, 2\}$ solves

$$\max_T \sum_{k=1}^{K} q_k\, u_{T(k),k} \quad \mathrm{s.t.} \quad \sum_{k=1}^{K} q_k\, \mathbf{1}[T(k) = 1] \leq b_{\mathrm{RAG}}. \tag{8}$$

This is the routing analogue of RD allocation: internal capacity sets $\widehat{d}_{0,k}$ via the cliff, and external retrieval is a second limited resource allocated under (8). In practice we solve the *empirical counterpart* of (8): rather than the entity population weighted by $q_k \propto f_k$, we optimize over the $N$ observed query instances, each weighted equally ($q_j = 1/N$), with frequent entities contributing proportionally more instances. The next result characterizes its optimal policy.

**Proposition 6.** *Assume $N$ query instances with uniform instance masses $q_j = 1/N$. For a query instance $j$, let the baseline (no-retrieval) utility be $u_{\mathrm{base},j} := \max\{u_{0,j}, u_{2,j}\}$, attained by the baseline action $T_{\mathrm{base}}(j) \in \arg\max\{u_{0,j}, u_{2,j}\}$ (generate or abstain, whichever scores higher). Let $\Delta_j := u_{1,j} - u_{\mathrm{base},j}$ be the gain from retrieving query $j$, for $1 \leq j \leq N$, and let the absolute retrieval budget be $M = \lfloor b_{\mathrm{RAG}} N \rfloor$. An optimal solution to the empirical counterpart of (8) is obtained by: (i) assigning every query $j$ to the baseline action $T_{\mathrm{base}}(j)$, and (ii) upgrading to retrieval the $M$ queries with the largest positive $\Delta_j$ (ties arbitrary). Equivalently, retrieval is assigned to the top-$M$ queries in $\Delta_j$ among those with $\Delta_j > 0$, and all other queries follow $T_{\mathrm{base}}$.*

*Proof.* Write the objective as baseline plus improvements. For any feasible empirical policy $T$, define the set of retrieved empirical queries $S(T) := \{j : T(j) = 1\}$. Relative to the baseline policy $T_{\text{base}}$, the objective difference is

$$\sum_{j=1}^{N} q_j\, u_{T(j),j} \;-\; \sum_{j=1}^{N} q_j\, u_{\text{base},j} \;=\; \sum_{j \in S(T)} q_j\,(u_{1,j} - u_{\text{base},j}) \;=\; \sum_{j \in S(T)} q_j\, \Delta_j,$$

since queries not in $S(T)$ contribute exactly $u_{\text{base},j}$. With uniform masses $q_j = 1/N$, the constraint $\sum_{j=1}^{N} q_j\, \mathbf{1}[T(j) = 1] \leq b_{\text{RAG}}$ is equivalent to $|S(T)| \leq M$. Thus, maximizing the improvement under a cardinality constraint is solved by taking the $M$ largest gains $\Delta_j$, excluding any with $\Delta_j \leq 0$. $\square$

This makes the RD link explicit: since $u_{0,k}$ depends on $\widehat{d}_{0,k}$, which the cliff (6) governs, the upgrade score $\Delta_k$ inherits a frequency geometry centered at $f_{\text{crit}}$, and retrieval is most valuable below the cliff, modulated by the relation-conditioned residual risk $\widehat{d}_1(R_k)$.

This raises a natural question: how close to optimal is the frequency score as a *discriminator* for hallucination risk? The next proposition gives a tight answer.

**Proposition 7** (Bayes-optimal pre-inference AUROC and converse bound). *Let $x = \log f \sim q$ on $[0, X]$ and $y \mid x \sim \mathrm{Bernoulli}(g(x))$ with $g(x) = a_{\min} + (a_{\max} - a_{\min})\,\sigma(\kappa(x - x_{\text{crit}}))$ monotone increasing. Let $\bar{a} = \mathbb{E}_q[g(X)]$ and $F_q(x) = \mathrm{Pr}_q(X \leq x)$.*

1. *The frequency score $s^*(x) = g(x)$ (equivalently $s^*(x) = x$) is the Bayes-optimal pre-inference ranking rule: among all measurable functions $s(x)$ of the query, it maximizes AUROC. The optimal value is*

$$\mathrm{AUROC}^* \;=\; \frac{1}{2} + \frac{\mathrm{Cov}_q\big(g(X),\, F_q(X)\big)}{\bar{a}(1 - \bar{a})}. \tag{9}$$

2. ***Converse.*** *No pre-inference predictor measurable with respect to entity frequency $f$ can exceed $\mathrm{AUROC}^*$; the cliff's steepness $\kappa$ therefore imposes a hard ceiling on* any *frequency-based hallucination detector, no matter how it is constructed. This ceiling is not a property of a particular score but of the information frequency carries about correctness. It depends on $\kappa$ through (9): $\mathrm{AUROC}^*$ is strictly increasing in $\kappa$, with*

$$\lim_{\kappa \to 0} \mathrm{AUROC}^* = \tfrac{1}{2}, \qquad \lim_{\kappa \to \infty} \mathrm{AUROC}^* = \frac{1}{2} + \frac{1}{2}\frac{\mathrm{Pr}_q(X < x_{\text{crit}})\,\mathrm{Pr}_q(X > x_{\text{crit}})(a_{\max} - a_{\min})}{\bar{a}(1 - \bar{a})}, \tag{10}$$

*so shallow cliffs ($\kappa \approx 0$) impose a hard discrimination ceiling that approaches chance, and steep cliffs ($\kappa \to \infty$) approach a theoretical maximum determined by the intrinsic noise of the capacity limits ($a_{\min}$ and $a_{\max}$).*

3. *For $q$ uniform on $[0, X]$ and $\kappa X \gg 1$:*

$$\mathrm{AUROC}^* \;=\; \frac{1}{2} + \frac{\frac{1}{X^2}\int_0^X x\, g(x)\, dx \;-\; \bar{a}/2}{\bar{a}(1 - \bar{a})}. \tag{11}$$

*Proof.* See Section A. $\square$

The argument is short: by the Neyman–Pearson lemma the likelihood ratio $g(x)/(1 - g(x))$ is the optimal test statistic, and since it is monotone in $x$, the frequency score induces the same ranking and is optimal; the closed form then follows from the Wilcoxon–Mann–Whitney identity, and monotonicity in $\kappa$ from a first-order stochastic-dominance argument on the class-conditional densities (Section A).

**Remark 5.** *The ceiling (9) sets the limit of frequency-based discrimination available before any model invocation. For Mistral-7B it is $\mathrm{AUROC}^* = 0.836$, which the frequency sigmoid (0.810) nearly attains. That the relation-aware `Freq + relation` score reaches 0.842, just above this frequency-only ceiling, is exactly what the converse permits: relation type is information beyond raw frequency, so a predictor using it is not bound by the frequency-only limit.*

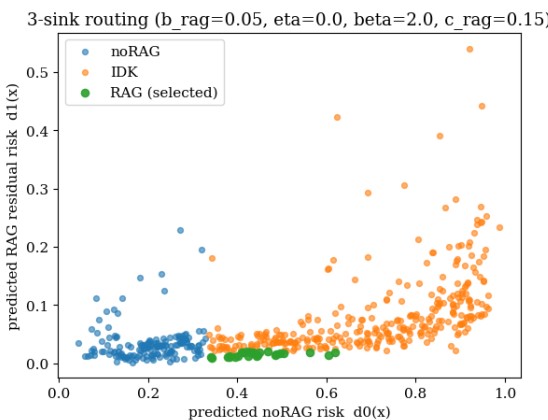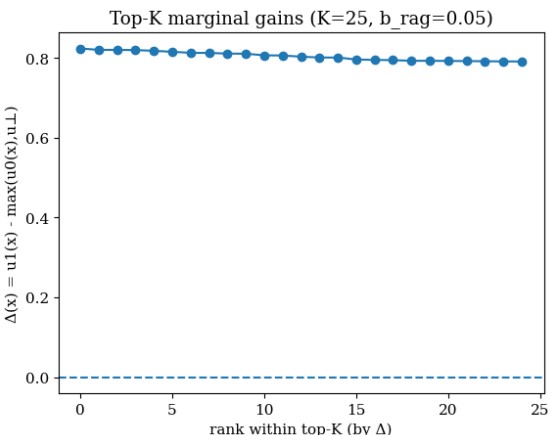

Figure 1: **Routing geometry. Left:** predicted noRAG distortion $\widehat{d}_{0,k}$ (6) versus predicted RAG residual distortion $\widehat{d}_{1,k}$ (7), colored by the three-action decision ($b_{\mathrm{RAG}} = 0.05$, $\beta = 2.0$, $c_{\mathrm{RAG}} = 0.15$). RAG is chosen only when noRAG risk is high and residual RAG risk low; IDK (abstain) dominates when both are high. **Right:** sorted top-$K$ marginal gains $\Delta_k$ (Proposition 6) among the top $K = 25$ upgrades ($b_{\mathrm{RAG}} = 0.05$); $\min \Delta_k > 0$ certifies the retrieval cap is binding and every upgrade is utility-improving.

To verify the result numerically, we evaluate the ceiling on Mistral-7B. Here $q$ is the distribution of log-frequencies $x = \log f$ across queries, supported on $[0, 7]$ (the empirical PopQA range); we take $q$ uniform as a tractable approximation, noting (9) holds for any $q$. Under uniform $q$, (11) gives $\mathrm{AUROC}^* = 0.816$, against an empirical frequency-sigmoid $\mathrm{AUROC} = 0.810$ (Table 5). Both use the same fitted $\kappa$, so this in-sample comparison is a *consistency check*, not independent validation. To validate out-of-sample, we estimate $\kappa$ (and the ceiling) on a training split and evaluate the empirical AUROC on a disjoint test split: the held-out ceiling is $0.822 \pm 0.010$ and the held-out AUROC $0.819 \pm 0.028$, so the near-saturation is not an in-sample artifact. The frequency sigmoid is by construction the optimal frequency-only predictor, so this tests whether the fitted cliff is well-specified and generalizes; that frequency is strong in absolute terms is shown separately by the held-out AUROC against the 0.5 chance baseline. The ceiling binds *frequency-only* predictors; signals using relation type or model confidence can and do exceed it (Section 6.4). Finally, Falcon-7B yields a higher ceiling ($\mathrm{AUROC}^* = 0.837$) despite a shallower cliff. This does not contradict the monotonicity of Proposition 7, which holds at fixed other parameters, whereas across models all parameters differ: holding Falcon's ($x_{\mathrm{crit}}, a_{\min}, a_{\max}$) fixed and varying $\kappa$ alone, $\mathrm{AUROC}^*$ does increase strictly, as the proposition requires; across the two models, Falcon's larger floor-to-ceiling gap outweighs its shallower slope.

Before turning to extensive experiments, we connect this routing formulation to error exponents and the rate–distortion perspective on sharp generalization transitions.

### 4.1 Reliability Exponents at High Frequency

The rate–distortion view tells us *where* reliable recall becomes possible, at the critical exposure $f_{\mathrm{crit}}$, but it is silent on what happens just past that boundary: once an entity is frequent enough to be storable, how quickly does error actually fall as it becomes more frequent still? This is the practitioner's question, and it turns out the sigmoid we already fit answers it: the decay rate above the cliff is the same steepness $\kappa$.

Write $x := \log f$ and $x_{\mathrm{crit}} := \log f_{\mathrm{crit}}$, and restrict attention to the head regime $x \geq x_{\mathrm{crit}}$, where the model is in principle capable of storing the fact. We say the head has reliability exponent $\eta \geq 0$ if excess error decays as $p_{\mathrm{exc}}(x) \approx \exp(\alpha - \eta(x - x_{\mathrm{crit}}))$, i.e. if log-error falls off linearly above the cliff with slope $-\eta$. That such a decay exists is model-independent: under any mechanism in which repeated exposure gives repeated chances to encode a fact, error must fall geometrically (if an entity at $x$ gets $m(x) = m_0 + c(x - x_{\mathrm{crit}})$ encoding opportunities, each landing the fact with probability $\rho$, a union bound gives $p_{\mathrm{err}}(x) \leq \epsilon_0 + (1 - \rho)^{m(x)}$, the exponential form above with $\eta = -c \log(1 - \rho) > 0$). But that argument only guarantees the rate is positive; what fixes it is

the sigmoid, whose upper tail is geometric with rate $\kappa$: for $x \geq x_{\mathrm{crit}}$, $1 - \sigma(\kappa(x - x_{\mathrm{crit}})) \sim \exp(-\kappa(x - x_{\mathrm{crit}}))$, so $p_{\mathrm{exc}}(x) \approx (a_{\max} - a_{\min}) \exp(-\kappa(x - x_{\mathrm{crit}}))$ and the exponent is simply $\eta = \kappa$.

We take $\kappa$ as the primary estimate rather than fitting $\eta$ afresh, since it comes from all 500 observations (Table 1) while a direct head-region fit sees only the tail: for Mistral-7B, $\kappa = 4.87$, so each decade past the cliff cuts head excess error by roughly $\exp(-\kappa) \approx 0.008$. A direct unbinned-likelihood fit on the $n = 199$ head examples ($x \geq x_{\mathrm{crit}} = 4.138$) returns a much smaller $\hat{\eta} \approx 0.64$ (bootstrap $[0.07, 1.20]$), but this is a feature of the data: real error bottoms out at an irreducible floor $\epsilon_0 \approx 0.21$, and a floorless fit reads that plateau as a gentle slope, deflating the exponent. The exponent also varies by relation, as the cliff location does, from $\eta \approx 0.6$ for `capital-of` to $\eta \approx 4.5$ for `screenwriter` (via the relation-specific $\kappa_g$): relation structure enters twice, setting both *where* the cliff falls and *how steeply* reliability climbs past it. That this direct fit is so uncertain raises a question we have left implicit, how much labeled data it takes to pin down the cliff at all, which the next section answers in terms of $\kappa$ and the query distribution.

## 5 Sample Complexity of Cliff Localization

A practical question follows from the estimation framework: how many labeled examples are required to localize $x_{\mathrm{crit}}$ to a given precision $\varepsilon$? This determines the minimum calibration cost for deploying the pre-inference risk score in a new domain, and explains the empirical finding that ECE $< 0.10$ is achievable with $\sim$75 calibration samples.

**Proposition 8** (Sample complexity of cliff localization)**.** *Let $x_1, \ldots, x_n \sim p(x)$ be i.i.d. log-frequencies on $[0, X]$ with binary labels $y_i \mid x_i \sim \mathrm{Bernoulli}(\sigma(\kappa(x_i - x_{\mathrm{crit}})))$. The maximum likelihood estimator $\hat{x}_{\mathrm{crit}}$ possesses the following Fisher information properties:*

    *(i) **Pointwise Information.** Let $\sigma'$ be the derivative of the logistic sigmoid. A single observation at $x$ contributes $\mathcal{I}(x_{\mathrm{crit}}; x) = \kappa^2 \sigma'(\kappa(x - x_{\mathrm{crit}}))$, which peaks at $\kappa^2/4$ at the cliff and decays exponentially away from it.*

    *(ii) **Zipf-1 Sampling** ($\alpha = 1$)**.** For $p(x) = 1/X$, the total information is $\mathcal{I}_n(x_{\mathrm{crit}}) \approx n\kappa/X$ (assuming $\kappa X \gg 1$). To achieve standard error $\mathrm{SE}(\hat{x}_{\mathrm{crit}}) \leq \varepsilon$, the required sample size is $n \geq X/(\kappa\varepsilon^2)$. Complexity is independent of $x_{\mathrm{crit}}$.*

    *(iii) **General Zipfian** ($\alpha \neq 1$)**.** For $p(x) \propto 10^{(1-\alpha)x}$ with normalization constant $Z_\alpha$, the total information and corresponding sample complexity bound are:*

$$\mathcal{I}_n(x_{\mathrm{crit}}) \approx \frac{n\kappa}{Z_\alpha} f_{\mathrm{crit}}^{1-\alpha} I(\kappa, \alpha), \qquad n \geq \frac{Z_\alpha}{\kappa \, \varepsilon^2 \, f_{\mathrm{crit}}^{1-\alpha} \, I(\kappa, \alpha)} \tag{12}$$

    *where $I(\kappa, \alpha) = \pi t / \sin(\pi t)$ for $t = (1 - \alpha) \ln(10)/\kappa \in (-1, 1)$. For steeper Zipfian distributions ($\alpha > 1$), $f_{\mathrm{crit}}^{1-\alpha}$ decreases with $f_{\mathrm{crit}}$, meaning tail cliffs require exponentially more samples to localize because natural sampling heavily underrepresents the informative transition window.*

*Proof.* See Section A. $\square$

**Numerical validation.** For Mistral-7B ($\kappa = 4.87$, $X = 7$), a target standard error of $\varepsilon = 0.06$ decades requires $n \approx X/(\kappa\varepsilon^2) \approx 399$ under natural Zipfian sampling, consistent with the empirical SE of 0.06 at $n = 500$; the modest gap reflects our stratified oversampling of the extreme tail, where Fisher information is exponentially suppressed. For the 75-sample ECE result, the theoretical SE is $\sqrt{X/(75\kappa)} \approx 0.138$ decades, and a first-order bound using the peak slope at the cliff ($a'(x) = (a_{\max} - a_{\min})\kappa/4$) gives a worst-case probability deviation $\approx (a_{\max} - a_{\min})\kappa \, \mathrm{SE}/4 \approx 0.109$. Since this is the worst case at the cliff boundary, the average over the query distribution is lower, consistent with the observed ECE $< 0.10$.

**Remark 6.** *The $1/\kappa$ dependence in (iii) means that steep cliffs are disproportionately easy to localize: doubling $\kappa$ halves the required sample size. This is because a sharp cliff concentrates Fisher information in a narrow window around $x_{\mathrm{crit}}$, making each nearby sample highly informative. Conversely, shallow cliffs (low $\kappa$, as in Falcon-7B with $\kappa = 2.10$) require approximately $4.87/2.10 \approx 2.3\times$ more labeled examples to localize to the same precision.*

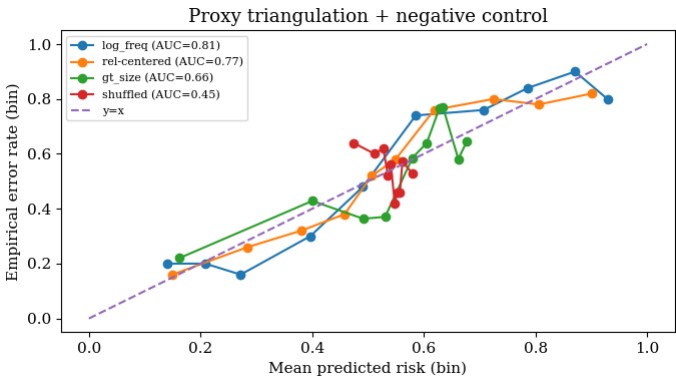

Figure 2: **Proxy triangulation and negative control.** Calibration-by-decile curves for pre-inference risk proxies. The shuffled negative control collapses into a tight cluster with no discriminative power. In contrast, the exposure-based proxies span a wide dynamic range and closely track the ideal $y = x$ calibration line, confirming their predictive reliability.

## 6  Experiments

**Dataset.**  We use PopQA (Mallen et al., 2023), a factual QA benchmark where each question concerns a single entity with known Wikipedia page view count, which we use as an exposure proxy for entity frequency in the training corpus. We evaluate on 500 stratified queries spanning four frequency bins: <1K, 1K–10K, 10K–100K, and >100K monthly page views. Our theory requires this proxy to track training exposure monotonically; we examine how well it does in Section 6.1 using multiple independent pre-inference proxies and a negative control.

**Models and Protocol**  We evaluate three model families: Mistral-7B-Instruct-v0.3, Falcon-7B-Instruct, and Qwen-2.5-7B-Instruct. Each query is evaluated in two modes: direct generation (no RAG) and retrieval-augmented generation. We record the generated answer, correctness (exact match against ground truth aliases), mean token probability (confidence), and token-level entropy.

### 6.1  Proxy Triangulation and Negative Controls

A key experimental assumption is that Wikipedia page views are approximately monotone in true training exposure. Popularity (page views) is what we measure, training exposure is what the theory concerns, and the two are linked through entity prominence but are not identical. The experiments below test how well the measured proxy tracks reliability; the open-corpus analysis in Section 6.5 examines the prominence–exposure link directly. To stress-test the proxy, we compare multiple *pre-inference* proxies and a negative control on the same evaluation set. Specifically, we evaluate: (i) our primary log-frequency risk $\hat{d}_k$ from Equation (4), (ii) a relation-centered log-frequency proxy (subtracting the per-relation median $\log f$ to control for cross-relation difficulty), (iii) an independent structural proxy based on ground-truth alias set size (measuring answer ambiguity), and (iv) a shuffled-frequency negative control.

Figure 2 plots the calibration-by-decile curves (mean predicted risk vs. empirical error rate) for each proxy. The shuffled control (red) completely breaks the signal; sorting by random values destroys alignment with correctness, collapsing the deciles into a non-discriminative cluster around the base error rate (AUC=0.45). In contrast, the genuine frequency proxies closely track the ideal $y = x$ calibration line across a wide range. Furthermore, the strong performance of the relation-centered proxy (AUC=0.77) confirms that the frequency signal persists after controlling for cross-relation difficulty. Together, these results show that our modeled risk probabilities track empirical error rates, supporting page views as a useful (though imperfect) proxy for exposure-linked prominence; the limits of this proxy are examined directly in Section 6.5.

## 6.2 Experiment 1: The Knowledge Cliff

We fit the sigmoid (Equation (4)) to binned accuracy-vs-frequency data using weighted nonlinear least squares with per-bin weights proportional to bin counts (i.e., $\sigma_i = 1/\sqrt{n_i}$). Table 1 reports the fitted parameters.

Table 1: Sigmoid fit parameters across models. All fits achieve weighted $R^2 > 0.89$, confirming the phase transition shape. $f_{\text{crit}}$ varies with training data: Mistral (better tail coverage) has a lower cliff.

| Model | Overall acc. | $f_{\text{crit}}$ | $\log(f_{\text{crit}})$ | $\kappa$ | $R_w^2$ |
|---|---|---|---|---|---|
| Mistral-7B | 0.462 | 12,726 | $4.10 \pm 0.06$ | 4.87 | 0.969 |
| Falcon-7B | 0.308 | 66,009 | $4.82 \pm 0.38$ | 2.10 | 0.892 |
| Qwen-2.5-7B | 0.326 | 50,843 | $4.71 \pm 0.18$ | 3.02 | 0.920 |

The key observation from Table 1 is the clustering: Falcon and Qwen share similar cliff locations ($f_{\text{crit}} \approx$ 51–66K), while Mistral's cliff is substantially lower ($f_{\text{crit}} \approx$ 13K). This is consistent with Proposition 5: the cliff reflects effective pretraining coverage, and in our evaluation Mistral shows higher accuracy in the 1K–10K transition range (0.248 vs. $\sim 0.14$ for Falcon/Qwen Table 2), consistent with better coverage of medium-frequency entities in the underlying training data.

Table 2: Per-stratum accuracy across models. All models show the same pattern: near-random performance in the tail (<1K) rising to 66–83% in the head (>100K). Mistral's advantage concentrates in the 1K–100K transition zone.

| Stratum | Mistral-7B | Falcon-7B | Qwen-2.5-7B |
|---|---|---|---|
| <1K | 0.144 | 0.104 | 0.136 |
| 1K–10K | 0.248 | 0.136 | 0.136 |
| 10K–100K | 0.624 | 0.336 | 0.312 |
| >100K | 0.832 | 0.656 | 0.720 |
| Overall | 0.462 | 0.308 | 0.326 |

**A note on $f_{\text{crit}}$ reporting conventions.** Because $f_{\text{crit}}$ is estimated by fitting a sigmoid, its value depends on the evaluation protocol (which queries, which prompt template, which binning). We report several $f_{\text{crit}}$ values for Mistral and Qwen under different protocols, and to avoid confusion we collect them in Table 3. Two distinctions matter. First, the *evaluation set*: our main experiments use a 500-query stratified set ($f_{\text{crit}} = 12,726$ for Mistral), while the fine-tuning study (Section 6.3) uses a larger held-out split (4,281 queries) on which the base model's cliff is 24,802, and the per-relation individual-prediction analysis (Section 6.6) uses an 8-bin global fit yielding $f_{\text{crit}} = 13,311$. Second, and more consequentially, the *prompt template*: for Qwen-2.5-7B the generic shared template used in Table 1 gives $f_{\text{crit}} = 50,843$, whereas model-native ChatML formatting (used for fair cross-size comparison in Section 6.7) gives 24,444, a $\sim 2\times$ shift.

Table 3: Canonical $f_{\text{crit}}$ values reported in this paper and the protocol producing each. The sigmoid *shape* is stable across protocols; the fitted *location* depends on evaluation set, binning, and prompt template, as expected for a fitted threshold.

| Value | Model | Protocol |
|---|---|---|
| 12,726 | Mistral-7B | 500-query set, generic template (main, Table 1) |
| 13,311 | Mistral-7B | 500-query set, 8-bin per-relation fit (Section 6.6) |
| 24,802 | Mistral-7B | 4,281-query held-out split (fine-tuning base, Table 4) |
| 50,843 | Qwen-2.5-7B | 500-query set, generic shared template (Table 1) |
| 24,444 | Qwen-2.5-7B | 500-query set, native ChatML template (Section 6.7) |

**Prompt-template sensitivity.** The Qwen 50,843 → 24,444 shift under ChatML formatting deserves comment, because it bears on our interpretation of $f_{\mathrm{crit}}$ as a model "fingerprint." Native formatting roughly halves the apparent cliff, i.e., the model reliably recalls entities $\sim 2\times$ rarer when prompted in its instruction-tuned format. A natural reading is that better prompting acts like additional effective exposure, shifting the cliff left. The practical implication is that $f_{\mathrm{crit}}$ is a property of the *model-plus-protocol*, not the weights alone; cross-model comparisons must therefore fix the template (as we do in the capacity-scaling experiment via `apply_chat_template`). We accordingly temper the "fingerprint" language: the sigmoidal *shape* and the *ordering* of models are stable, but the absolute cliff location should always be reported with its eliciting protocol. This template dependence does not affect the within-protocol results (risk scoring, routing, scaling), all of which hold the template fixed.

### 6.3 Experiment 2: Fine-Tuning Does Not Improve the Cliff

If the knowledge cliff is fundamentally constrained by pretraining capacity allocation (Proposition 5), then lightweight post-training should not be able to improve $f_{\mathrm{crit}}$ (shift the cliff left, so that rarer entities become reliably recalled): such adaptation perturbs the pretrained weights but cannot add capacity or reorganize how bits were allocated, so at most it should leave the cliff in place and, through forgetting, may smear it. To test this, we fine-tune Mistral-7B with QLoRA on a 9,987-question training split of PopQA using three weighting schemes: uniform, square-root-inverse frequency, and inverse frequency (a theoretically motivated reweighting from the R-D framework). Table 4 reports the results on a held-out evaluation split (4,281 questions); note that the base model $f_{\mathrm{crit}} = 24,802$ differs from Table 1 due to this specific larger split.

Table 4: Fine-tuning reshapes individual predictions but blurs the cliff boundary. No weighting scheme shifts the cliff to the left (improves recall). Instead, FT increases the point estimate of $f_{\mathrm{crit}}$ and noticeably degrades the sigmoid goodness-of-fit $(R_w^2)$, indicating a smearing of the pretraining capacity boundary.

| Weighting | $f_{\mathbf{crit}}$ | $\log(f_{\mathbf{crit}}) \pm \mathbf{SE}$ | $R_w^2$ |
|---|---|---|---|
| Base (no FT) | 24,802 | $4.39 \pm 0.17$ | 0.938 |
| Uniform FT | 45,999 | $4.66 \pm 0.33$ | 0.864 |
| $\sqrt{\mathrm{inv}}$-freq FT | 63,620 | $4.80 \pm 0.32$ | 0.846 |
| Inv-freq FT | 48,709 | $4.69 \pm 0.23$ | 0.842 |

Lightweight adaptation does not reorganize the capacity allocation. Fine-tuning does change individual predictions (McNemar $p = 0.0072$), but through format adaptation and catastrophic forgetting, not knowledge acquisition: inverse-frequency FT fixes 266 tail predictions (<1K) by teaching the QA format while breaking 143 confident head predictions (10K–100K). It does not improve $f_{\mathrm{crit}}$: the point estimates drift right and their uncertainty grows, leaving the shift statistically indistinguishable from the base cliff ($z = 1.03$, $p > 0.05$), and the $R_w^2$ drop across all variants indicates the pretrained boundary becomes "smeared." This evidence concerns *lightweight QLoRA adaptation on in-distribution QA data*; continued pretraining or full fine-tuning at larger scale is left to future work. Within this scope, Proposition 5 is supported: lightweight downstream adaptation does not improve the pretrained cliff, and tends to smear it.

### 6.4 Experiment 3: Hallucination Prediction via Frequency

We evaluate the sigmoid risk score $\hat{d}_k = 1 - \sigma(\kappa(\log f_k - \log f_{\mathrm{crit}}))$ as a hallucination detector on the **500-query PopQA / Mistral-7B set**, with hallucination (an incorrect direct-generation answer) as the positive class. We compare it against five baselines: *raw model confidence* ($1-$ mean token probability); *P(True)* (Kadavath et al., 2022), which shows the model its own answer and reads $1 - P(\mathrm{True})$ from the option-token logits (two forward passes); *semantic entropy* (Kuhn et al., 2023), entropy over semantic clusters of ten temperature-1 samples in the discrete short-answer variant (Farquhar et al., 2024) (eleven passes); a *regime-aware joint score* that leans on confidence mainly above the cliff, where it is more informative; and a *combined* 5-fold out-of-fold logistic model over frequency, confidence, minimum token probability, and entropy.

**Results.**  The striking result (Table 5) is that the frequency sigmoid reaches AUROC = 0.810 while running nothing: it beats not only raw confidence (0.772) but the far more expensive P(True) (0.714) and semantic entropy (0.733), which cost two and eleven forward passes. Adding relation structure, still with no model execution, lifts it further to 0.842 (`Freq + relation`). The self-consistency methods fail here for a reason our framework anticipates: at low frequency the model reverts to the relation prior and generates *consistent but wrong* answers, so sampling-based uncertainty looks lowest exactly where hallucination risk is highest. Confidence does carry information, but mostly above the cliff: folding it in through the regime-aware score lifts discrimination to 0.835, and the full combination of frequency and inference-time uncertainty does best (AUROC = 0.856, AUPRC = 0.873, Brier = 0.153): the two signals are complementary, not redundant.

Table 5: Hallucination detection metrics on the 500-query PopQA/Mistral-7B evaluation set. The frequency sigmoid is the only score available with *no model execution* (zero forward passes) and achieves the best AUROC, ECE, and Brier among all single-signal baselines, including the stronger inference-time estimators P(True) and semantic entropy. "Passes" counts total model forward passes per query, including the answer generation itself.

| Method | Passes | AUROC | AUPRC | ECE | Brier |
|---|---|---|---|---|---|
| Freq sigmoid | 0 | 0.810 | 0.809 | 0.041 | 0.169 |
| Freq + relation | 0 | 0.842 | – | – | – |
| Model confidence (raw) | 1 | 0.772 | 0.785 | 0.414 | 0.384 |
| P(True) (Kadavath et al., 2022) | 2 | 0.714 | – | 0.321 | 0.319 |
| Semantic entropy (Kuhn et al., 2023) | 11 | 0.733 | – | 0.347 | 0.332 |
| Regime-aware joint score | 1 | 0.835 | 0.812 | – | – |
| Freq + conf (combined) | 1 | 0.856 | 0.873 | 0.033 | 0.153 |

**Calibration and aggregation.**  Raw confidence is badly calibrated (ECE = 0.414), and post-hoc fixes (temperature, Platt, isotonic) cut ECE sharply (best 0.021) only by sacrificing discrimination, the familiar calibration–discrimination tradeoff. Even after those fixes, frequency remains the strongest single feature on both AUROC (0.810 vs. 0.775) and Brier (0.169 vs. 0.192–0.200). The reason frequency wins globally while confidence wins *within* a stratum (e.g. 0.769 above 100K) is a Simpson's-paradox effect: exposure decides whether a fact is stored at all, so frequency captures coverage across strata, while confidence only captures generation-time variability within them. The two are therefore complementary, and combining them gives the best calibration and discrimination together (AUROC = 0.856, Brier = 0.153; Figure 3, bottom-right).

**Transferability and sample efficiency.**  The detector holds up under resampling, distribution shift, and scarce labels (Figure 5). Across 100 random 50/50 splits the cliff barely moves ($f_{\text{crit}}$ mean 12,988, CV = 0.23), and out-of-sample ECE stays roughly $6\times$ below raw confidence ($0.069 \pm 0.021$ vs. $0.415 \pm 0.022$). Leave-one-stratum-out transfer predicts a held-out stratum's hallucination rate to mean |error| = 0.060, worst in the saturated tail (>100K: 0.083), and calibration quality climbs quickly with only a handful of labels. That confidence's correct-vs-incorrect gap stays nearly flat across strata ($\Delta \approx 0.075$) is the same story from another angle: confidence discriminates within a stratum, frequency supplies the coverage across them.

**Routing gains from pre-inference structure.**  The same signal makes a strong retrieval router. Our best *pre-inference* policy, `Freq + relation`, reaches 0.610 accuracy at a 20% RAG budget, close to the oracle[1] ceiling (0.642) and matching every confidence-based policy without model execution (raw confidence 0.608, Platt-calibrated 0.600). Adding post-inference confidence (`Freq + conf combined`) helps only marginally at 20% (0.622 vs. 0.610), and the edge disappears by 40%, where the purely pre-inference policy draws level with it (0.762 vs. 0.760); to hit 80% accuracy, every frequency-based policy needs the same 47% budget, against 54% for confidence and 72% for random. Because `Freq + relation` costs no model execution, this says something concrete about the threshold-routing view of Section 4: nearly all the routing utility is available before a single token is generated. Figure 4 shows the frontiers.

---

[1]Oracle routing uses per-query counterfactual `noRAG`/`RAG` outcomes available only offline; it is not implementable at runtime.

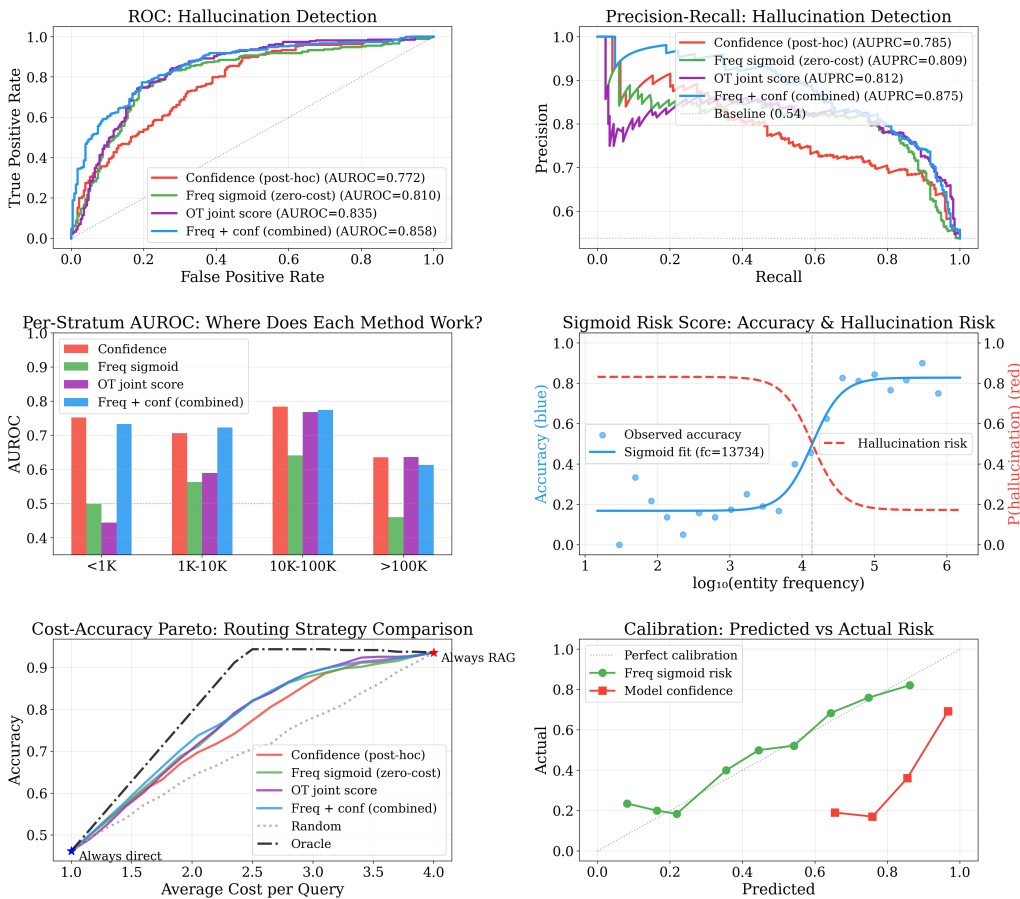

Figure 3: **Hallucination prediction results (Mistral-7B, 500-query PopQA).** Top: ROC and PR curves; the pre-inference frequency sigmoid performs well without an LLM forward pass, approaching post-hoc calibrated confidence. Middle: per-stratum AUROC and the fitted cliff ($f_{\text{crit}} = 13{,}734$ under this panel's binning; consistent with the canonical 12,726 of Table 1 up to bin choice). Frequency is the stronger signal across strata, confidence within them. Bottom: routing frontier and calibration; the combined frequency+confidence score is best overall (AUROC 0.856).

Table 6: **Generalization to unseen relations.** Proxy calibration is trained on a subset of relation types and evaluated on held-out relations; i.e., calibration trained on in-relations only; held-out relations are never seen in training. The shuffled-frequency negative control collapses to near-chance.

| Method | AUC (in-rel) ↑ | Brier (in-rel) ↓ | AUC (held-out rel) ↑ | Brier (held-out rel) ↓ |
|---|---|---|---|---|
| log_freq_sigmoid | 0.7839 | 0.1886 | **0.8444** | **0.1666** |
| struct_risk | **0.8667** | **0.1435** | 0.8108 | 0.1701 |
| 1-conf | 0.7792 | 0.2956 | 0.7613 | 0.4602 |
| shuffled_log_freq | 0.4503 | 0.2569 | 0.4419 | 0.2459 |

**Unseen-relation generalization.** Holding out whole relation types (Table 6), the log-frequency sigmoid still transfers strongly to unseen relations (held-out AUC = 0.844, Brier = 0.167), while the shuffled-frequency control collapses to chance (0.442). The held-out AUC slightly exceeding the in-relation one is not leakage but sampling variance: the frequency sigmoid has no relation-specific parameters, and indeed `struct_risk`, which does, shows the opposite (in-relation 0.867 → held-out 0.811), the direction a relation-aware model should move.

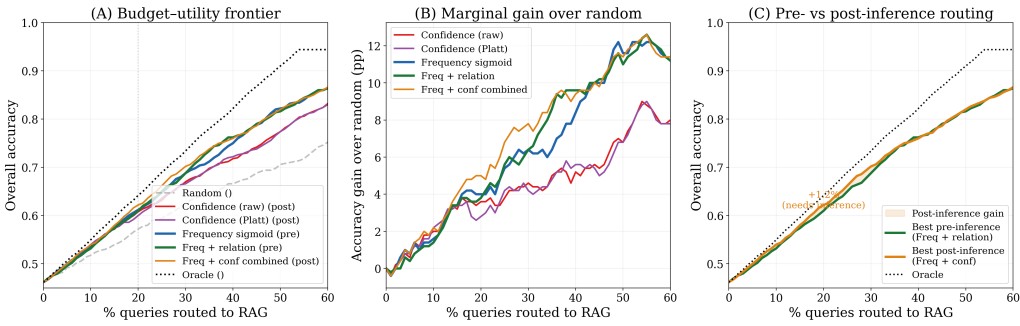

Figure 4: **Budget–utility frontier for retrieval routing.** For each RAG budget (% of queries routed to retrieval), queries are sorted by risk score and the highest-risk fraction is routed; accuracy uses recorded `no_rag_correct`/`rag_correct`. (A) Full frontier: the pre-inference `Freq + relation` policy (green) stays competitive with the confidence-based policies and approaches the oracle. (B) Gain over random routing: frequency-based policies improve most, with `Freq + conf combined` best at higher budgets. (C) Pre- vs. post-inference: at 20% budget `Freq + relation` reaches 0.610, against 0.622 for the best post-inference baseline (`Freq + conf`).

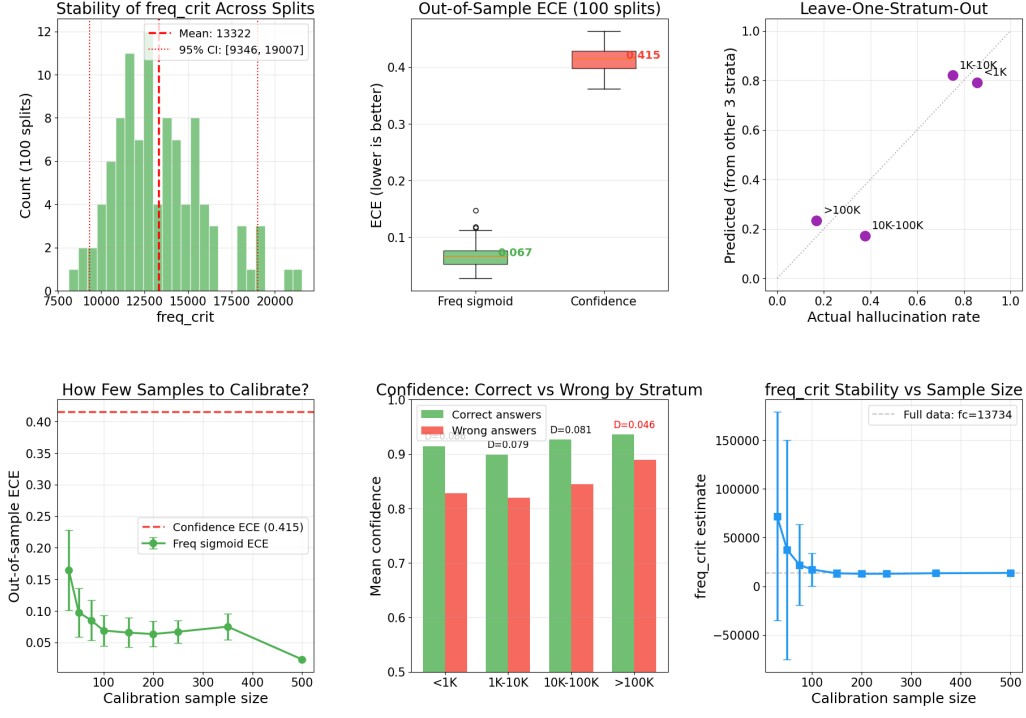

Figure 5: **Sigmoid transferability validation.** Top-left: the fitted threshold $f_{\text{crit}}$ across 100 random 50/50 train/test splits. Top-center: out-of-sample ECE. Top-right: LOSO transfer, comparing predicted vs. actual *stratum-level* hallucination rates. Bottom-left: sample efficiency for calibrating the sigmoid with limited labeled data. Bottom-center: mean confidence for correct vs. wrong answers by stratum showing a nearly constant separation. Bottom-right: $f_{\text{crit}}$ estimates stabilize as calibration sample size increases.

### 6.5 Experiment 4: Cross-Model Transfer

We evaluate cross-model transfer of the frequency-based sigmoid risk model on Falcon-7B and Qwen-2.5-7B using the same 500-query PopQA evaluation set as in the Mistral experiments. For each source model, we fit a sigmoid accuracy–frequency curve and convert it to a risk score; we then apply that fitted curve *without*

*refitting* to a target model's frequency-stratified outcomes and measure calibration error (ECE). Table 7 reports the resulting transfer matrix.

Table 7: Cross-model transfer ECE on the 500-query PopQA evaluation set. Diagonal entries (bold) are self-fit. Models with similar cliff locations (Falcon $\leftrightarrow$ Qwen) transfer especially well. Even worst-case sigmoid transfer (Mistral $\rightarrow$ others, ECE $\approx 0.154$) remains much better than raw confidence ECE (0.482–0.592).

| Fit on \ Predict on | Falcon-7B | Mistral-7B | Qwen-2.5-7B |
|---|---|---|---|
| Falcon-7B | **0.037** | 0.153 | 0.029 |
| Mistral-7B | 0.154 | **0.026** | 0.136 |
| Qwen-2.5-7B | 0.039 | 0.135 | **0.016** |
| Raw confidence ECE | 0.592 | 0.482 | 0.511 |

The transfer matrix reveals a clear block structure: Falcon $\leftrightarrow$ Qwen transfer ECEs (0.029–0.039) are close to self-fit performance, whereas transfer between Mistral and either Falcon or Qwen degrades to $\sim 0.154$. This pattern closely tracks the fitted cliff locations: Falcon and Qwen have similar critical frequencies ($f_{\text{crit}} \approx 51\text{K–66K}$), while Mistral's cliff is substantially lower ($f_{\text{crit}} \approx 13\text{K}$).

This variation in $f_{\text{crit}}$ is consistent with differences in pretraining data curation. One plausible explanation is that a corpus upsampling high-quality factual content raises a model's effective exposure to rare entities, so reliable recall arrives at a lower apparent page-view threshold; this has been suggested for more heavily curated corpora, but the training data here is not public, we cannot verify it directly, and other factors (tokenization, architecture, training recipe) may also contribute. The transfer results thus indicate that the *sigmoidal form* of the boundary is stable across LLMs, while the $f_{\text{crit}}$ location reflects the pretraining distribution. We use "fingerprint" loosely: per the prompt-template note (Table 3), the absolute location is protocol-dependent, so it fingerprints the model-plus-protocol, not the weights in isolation; the cross-model claims here hold the protocol fixed.

### 6.6 Experiment 5: From Population to Individual Prediction

Experiments 1–4 establish the frequency sigmoid as a strong, well-calibrated *population-level* risk estimate. But within a narrow stratum all queries have similar frequency and so similar frequency-based risk, leaving frequency with little *within-stratum* discrimination. Here we show that *relation structure* supplies the missing individual-level signal.

**Methodology.** Each PopQA question instantiates a semantic relation ("Who directed $X$?", "What is $X$ the capital of?"), which we recover by pattern matching. For each relation $r$ with enough support ($n \geq 15$) we fit its own sigmoid $\text{acc}_r(f) = a_{\min,r} + \frac{a_{\max,r} - a_{\min,r}}{1 + \exp(-\kappa_r(\log f - \log f_{\text{crit}}^{(r)}))}$, letting both the cliff $f_{\text{crit}}^{(r)}$ and slope $\kappa_r$ vary: if the per-relation cliffs differ substantially, relation type carries risk information beyond frequency alone. All prediction metrics are cross-validated and split into **pre-** and **post-inference** features. The fits (Table 8) show the cliff moving by $76\times$ across well-sampled relations, from `author` ($f_{\text{crit}} = 340$) to `director` (25,857); restricting to well-fit relations ($R^2 > 0.9$) still leaves a $27\times$ range (`country` 941 to `director`). The bootstrap CIs are wide, so the exact multiplier is uncertain, but the order-of-magnitude spread is not. (The 8-bin global fit here is $f_{\text{crit}} = 13{,}311$, $R^2 = 0.995$, matching the canonical 12,726 up to binning; we measure shifts against 12,726 for consistency.) The spread is consistent with relation-dependent differences in encoding opportunity and storage difficulty, though pinning down the cause would need corpus-level co-occurrence data.

**Within-stratum discrimination.** Evaluating AUROC separately inside each frequency band (Table 9) exposes a clean two-regime pattern. In the tail ($< 1\text{K}$) relation type is the strongest signal (0.779), ahead of confidence (0.752) and frequency-only risk (0.444). The low frequency-only number is expected by design: within a narrow band $f$ barely varies, so the frequency score cannot rank queries and dips below chance only through sampling noise. Higher up, confidence takes over (10K–100K: 0.784; $> 100\text{K}$: 0.636). The reading is

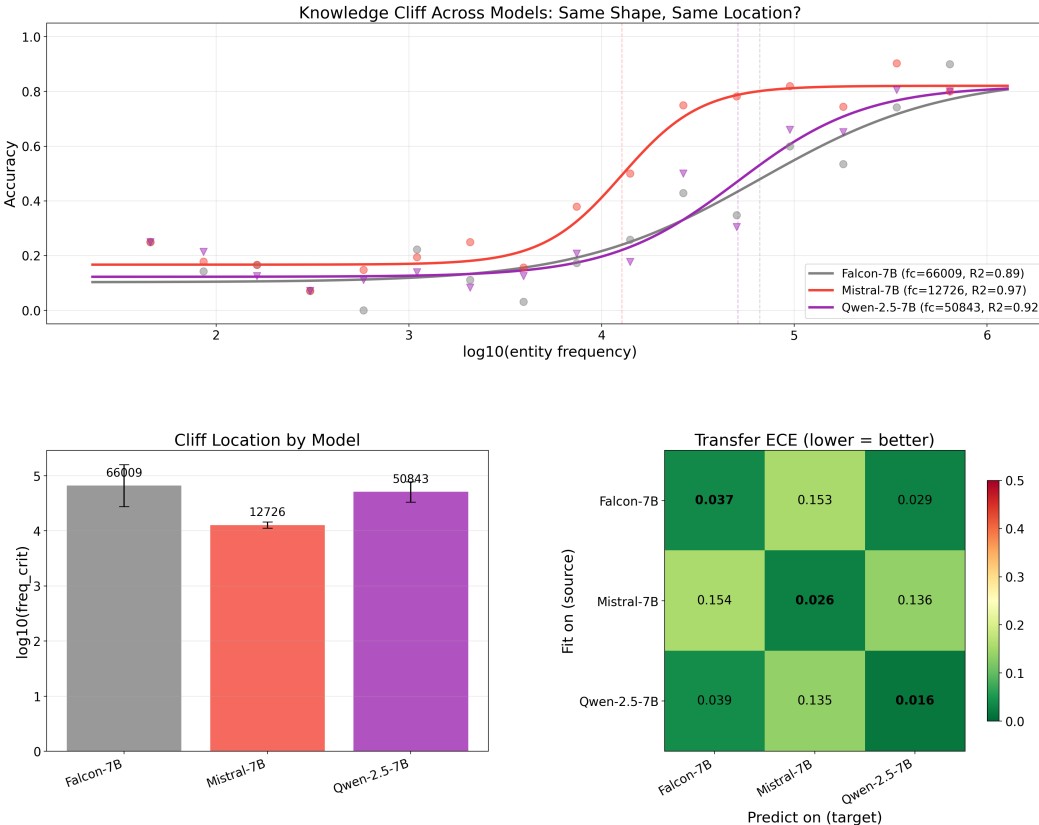

Figure 6: **Cross-model sigmoid transfer (500-query PopQA evaluation set).** Top: all three models exhibit sigmoidal transitions (weighted $R^2 > 0.89$). Mistral's cliff (red, $f_{\text{crit}} = 12{,}726$) is shifted left relative to Falcon (gray, 66,009) and Qwen (purple, 50,843). Bottom-left: fitted cliff location by model with 1-SE error bars. Bottom-right: transfer ECE heatmap; the Falcon/Qwen block transfers well (ECE 0.029–0.039), while Mistral-to-others transfer degrades (ECE ≈0.154), consistent with the larger shift in $f_{\text{crit}}$.

that the tail question is *which facts survive low exposure*, whereas in the head most facts are covered and the residual failures are generation-time variability, which confidence sees.

**Combined individual-level predictors.** Under 10-fold cross-validation, the pre-inference frequency score already beats post-inference confidence as a single feature ($0.813 \pm 0.050$ vs. $0.769 \pm 0.073$), and adding relational structure gives a strong *fully pre-inference* predictor ($0.842 \pm 0.055$). Folding in confidence reaches the global maximum ($0.875 \pm 0.044$), but that last +0.033 costs a full forward pass: most of the predictive power is available from query metadata before any generation.

**Generalization checks.** To check the relational signal (Figure 7) is portable rather than benchmark-specific, we run three tests. Leave-One-Relation-Out gives mean AUROC = 0.792 across 14 held-out relations. Replacing exact relation identities with generic structural features (alias count, median frequency, within-relation spread) gives $0.819 \pm 0.043$, about 30% of the identity-based gain over frequency alone. And Leave-$K$-Relations-Out ($K \in \{3, 5\}$) stays above 0.806, so the signal survives even with up to a third of relation types unseen.

### 6.7 Experiment 6: Capacity Scaling

We test Proposition 5 directly on the Qwen2.5-Instruct family (0.5B–14B). Because all five were trained on the same 18T-token corpus, parameter count is the only variable that moves. On the 500-query PopQA set

Table 8: Relation-specific sigmoid fits (Mistral-7B, PopQA; relations with $n \geq 15$). Point estimates of $f_{\text{crit}}$ span $76\times$; restricting to well-fit relations ($R^2 > 0.9$) gives a conservative $27\times$ range. 90% bootstrap CIs (2,000 resamples, degenerate rail-hitting fits discarded) are wide, especially for poorly-fit relations (`author`, `capital-of`), so the precise multiplier is uncertain while the order-of-magnitude heterogeneity is robust. $R^2$ is on binned accuracies; "Shift" is the log-distance from the global $f_{\text{crit}} = 12,726$.

| Relation | n | $f_{\text{crit}}$ | $R^2$ | 90% CI on $f_{\text{crit}}$ | Shift |
|---|---|---|---|---|---|
| author | 33 | 340 | 0.672 | [160, 267,842] | $-1.57$ (easier) |
| capital-of | 84 | 5,428 | 0.846 | [292, 1,811,503] | $-0.37$ |
| country | 31 | 941 | 0.961 | [59, 86,902] | $-1.13$ |
| composer | 34 | 13,394 | 0.999 | [2,547, 33,924] | $+0.03$ |
| genre | 51 | 13,355 | 1.000 | [2,429, 78,593] | $+0.03$ |
| producer | 46 | 12,615 | 0.925 | [7,029, 18,701] | 0.00 |
| screenwriter | 68 | 19,100 | 0.995 | [8,378, 27,676] | $+0.18$ |
| director | 66 | 25,857 | 0.965 | [8,009, 28,334] | $+0.31$ |
| **Global** | 500 | 12,726 | 0.969 | | , , |

Table 9: Within-stratum AUROC for hallucination prediction (Mistral-7B, PopQA). "Freq" denotes the frequency-sigmoid risk score. Relation-derived features are computed from held-in data only. Relation dominates in the tail ($< 1$K), while confidence dominates in higher-frequency strata. Frequency-only AUROC $\approx 0.5$ within strata is expected: frequency barely varies within a band, so it provides no within-stratum ranking signal (values below 0.5 are sampling noise).

| Stratum | n | Freq | Conf | Relation | Best |
|---|---|---|---|---|---|
| <1K | 125 | 0.444 | 0.752 | **0.779** | 0.779 |
| 1K–10K | 125 | 0.590 | **0.706** | 0.619 | 0.706 |
| 10K–100K | 125 | 0.660 | **0.784** | 0.514 | 0.784 |
| >100K | 125 | 0.510 | **0.636** | 0.424 | 0.636 |

the cliff follows a clean power law: $\log_{10} f_{\text{crit}} = 4.96 - 0.52 \log_{10} P$ (weighted $\hat{\alpha} = 0.52 \pm 0.06$, $R_w^2 = 0.97$; unweighted $R^2 = 0.94$; $n = 5$), with $f_{\text{crit}}$ falling monotonically from 161,169 at 0.5B to 23,940 at 14B. In practical terms, each decade of parameters buys about half a log-decade of frequency: a $10\times$ larger model reliably recalls entities roughly $3\times$ rarer.

**Robustness of the exponent.** With only five points and one degenerate fit (Qwen2.5-7B, whose $\kappa$ hits the optimizer bound so its cliff cannot be localized; we down-weight it via its large $\log f_{\text{crit}}$ SE), we stress-test the exponent four ways. Excluding the 7B point gives $\hat{\alpha} = 0.55 \pm 0.08$ ($n = 4$, $R_w^2 = 0.96$), indistinguishable from the full fit, so the result does not hinge on it. An unweighted OLS on all five gives $\hat{\alpha} = 0.58$ ($R^2 = 0.95$), bracketing the exponent in $[0.52, 0.58]$. Leave-one-model-out gives $\hat{\alpha} \in [0.42, 0.70]$ (widest when the 14B endpoint, the longest lever arm, is dropped), and a residual bootstrap ($B = 5000$) gives a 90% interval of $[0.48, 0.67]$. We therefore report the exponent as $\approx 0.5$ with this uncertainty rather than a precise constant: $f_{\text{crit}}$ falls with capacity at roughly half a frequency-decade per parameter-decade.

Scaling shifts the cliff but does not abolish the tail: accuracy below 1K page views stays near 10% at every model size (Figure 8, bottom-right), so retrieval remains necessary no matter how large the model. At matched parameter count, Mistral-7B sits 0.28 log-decades below Qwen-7B ($f_{\text{crit}} = 12,726$ vs. 24,444); this is a single-point cross-family observation, not a controlled comparison, but it is consistent with a smaller, more curated corpus encoding knowledge more efficiently per parameter Jiang et al. (2023a), shifting the intercept without changing the slope. (The Qwen-7B value here, 24,444, differs from the 50,843 of Table 1 because this experiment uses model-native ChatML formatting via `apply_chat_template` for fair cross-size comparison, whereas Experiment 1 used a shared generic template.)

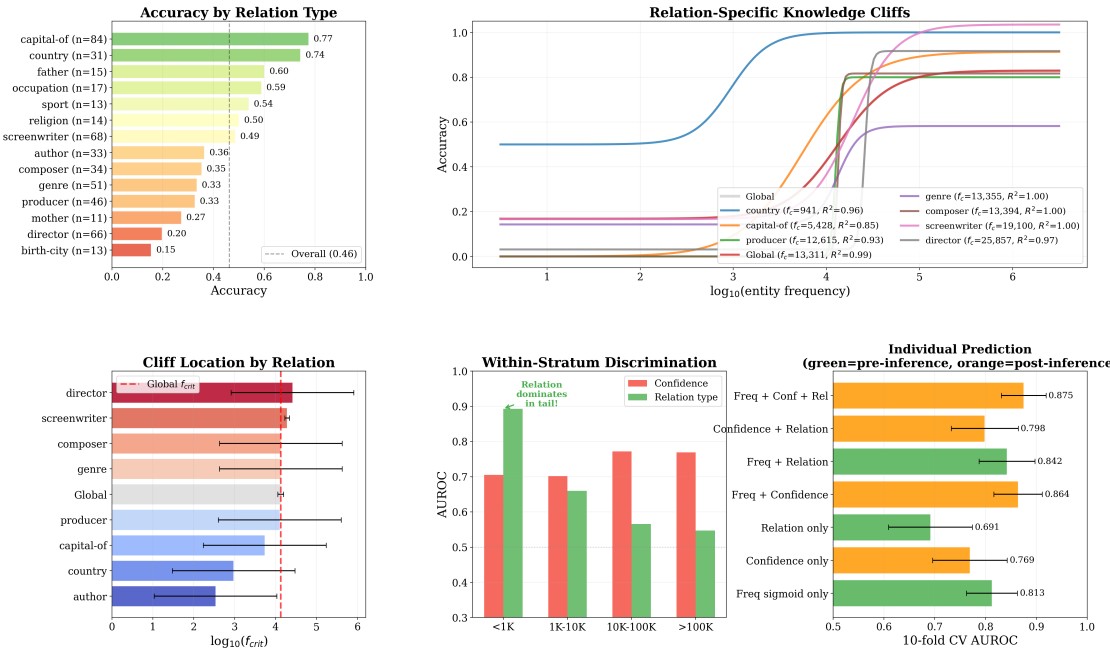

Figure 7: **Relation-specific knowledge cliffs and individual-level prediction (Mistral-7B).** Top row: accuracy and sigmoidal cliffs vary across relation types, the heterogeneity behind the global average. Bottom-left: critical frequencies ($f_{\mathrm{crit}}$) span more than an order of magnitude across relations (27–76×; bootstrap CIs in Table 8). Bottom-center: within-stratum discrimination, relation identity is the strongest signal in the rare-entity tail (<1K), confidence in the well-exposed head. Bottom-right: 10-fold CV; `Freq + relation` (green) is a competitive pre-inference routing baseline against post-inference combinations (orange).

## 7 Discussion

The rate–distortion framing does more than fit the sigmoid. The sigmoidal accuracy curve on its own is descriptive: one could fit it without any information-theoretic interpretation. The capacity-allocation view adds predictions the curve does not contain. Treating capacity as a finite resource allocated across a Zipfian source predicts how the critical frequency moves with model size ($f_{\mathrm{crit}} \propto P^{-\beta}$), a relationship in the parameter count $P$ that an accuracy-versus-frequency curve has no way to express. The same view yields a converse: an attainable ceiling $\mathrm{AUROC}^*$ on how well *any* frequency-only score can rank hallucinations, an information limit rather than a fit. And it explains why confidence becomes unreliable at low frequency: at zero allocated rate the output reverts to the relation prior regardless of correctness, so the overconfidence we observe on rare entities (Section 6.4) is what the view predicts. The assumptions behind these predictions (Section 3) are posited and empirically checked, not derived from a source coding theorem.

The routing and abstention results point the same way: a two-stage architecture for managing hallucinations. Because `Freq + relation` orders risk correctly both globally and within strata, it works as a cheap pre-inference triage step, with post-inference confidence held back for borderline cases. This ordering holds across the abstention utility thresholds we tried, which suggests query-side structure is a reliability signal that does not depend on the particular downstream intervention.

We would also like to remark on a converse limit that our abstention view shares with the "arbitrary facts" model of Kalai et al. (2025), in which generalization beyond memorization is information-theoretically impossible (for prompts whose labels are unsupported by training evidence, no learner beats random guessing in expectation). There the *missing mass* of prompts is a fundamental limit on selective utility. Let

$$M_0 \ := \ P\big(\{c \in \mathcal{C} : c \notin \{c^{(1)}, \ldots, c^{(N)}\}\}\big)$$

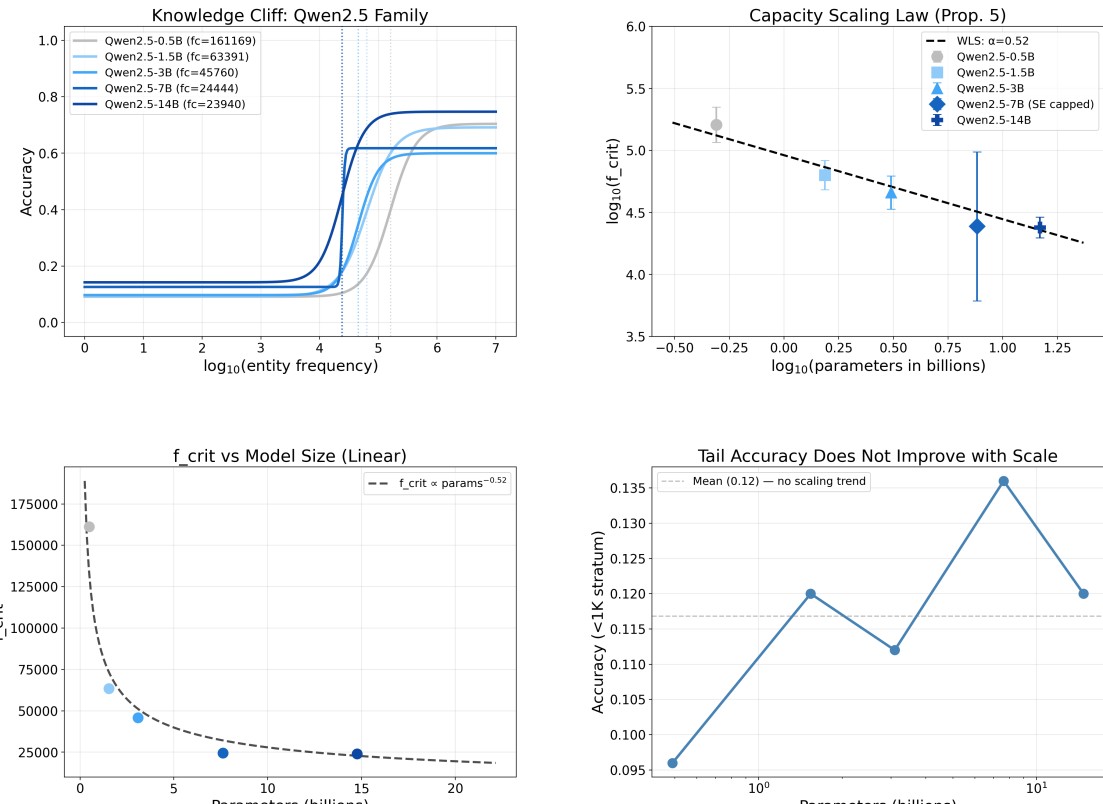

Figure 8: **Capacity scaling law (Experiment 6).** *Top-left:* Knowledge cliff curves for the Qwen2.5-Instruct family, all trained on the same 18T-token corpus. Larger models shift the cliff toward rarer entities. (Qwen2.5-7B is excluded from the visual curves due to a degenerate sigmoid fit, but included in regressions). *Top-right:* Log–log scaling law: $\log_{10} f_{\text{crit}} = 4.96 - 0.52 \log_{10} P$ ($\hat{\alpha} = 0.52 \pm 0.06$, $R_w^2 = 0.97$, $n = 5$; exponent robust to leave-one-out and bootstrap, $\hat{\alpha} \in [0.42, 0.70]$). Error bars show 1-SE uncertainty. *Bottom-left:* The same scaling law on linear axes, illustrating the power-law decay $f_{\text{crit}} \propto P^{-\beta}$ with $\hat{\beta} \approx 0.5$. *Bottom-right:* Tail accuracy (<1K stratum) vs. model size. Performance remains trapped near 10% across all scales; minor non-monotonic fluctuations are statistical noise driven by the small stratum size ($n = 125$), demonstrating that parameter scaling alone cannot rescue tail recall.

be the probability that a test prompt was never seen in training. Under arbitrary facts, any unseen prompt's output is independent of the true label, so expected correctness is at most $1/|\mathcal{Y}|$; if $u_{\text{abst}} \geq 1/|\mathcal{Y}|$, abstaining weakly dominates guessing there, and the utility loss from perfection is controlled by $M_0$.

**Corollary 9** (Missing-mass upper bound)**.** *Assume $u_{\text{abst}} \geq 1/|\mathcal{Y}|$. Then, conditional on any realized training sample $S$,*

$$\mathbb{E}[u \mid S] \ \leq \ 1 - (1 - u_{\text{abst}}) \, M_0(S),$$ (13)

*where the expectation is over the test prompt, the arbitrary-facts label assignment (for unseen prompts), and any predictor randomness.*

*Proof.* See Section B. □

A key point, consistent with Kalai et al. (2025), is that missing mass is also *estimable* from the sample via singleton statistics. Let $K_1$ be the number of prompts that appear exactly once in $\{c^{(1)}, \ldots, c^{(N)}\}$, and define the Good–Turing estimator $G_0 := K_1/N$. A classical result of McAllester & Schapire (2000)

shows that the missing mass concentrates around this estimator: for i.i.d. samples from a countable domain, $|M_0 - G_0| \leq \varepsilon_{\mathrm{GT}}(N,\delta)$ with probability at least $1-\delta$, where $\varepsilon_{\mathrm{GT}}(N,\delta) = O\big(\sqrt{\log(N/\delta)/N}\big)$.

Combining the missing-mass utility bound (13) with this concentration result yields a sample-dependent upper bound based entirely on the observable singleton fraction. Specifically, on the high-probability event $M_0(S) \geq \big(G_0 - \varepsilon_{\mathrm{GT}}(N,\delta)\big)_+$, we have:

$$\mathbb{E}[u \mid S] \ \leq \ 1 - (1 - u_{\mathrm{abst}})\, M_0(S) \ \leq \ 1 - (1 - u_{\mathrm{abst}})\big(G_0 - \varepsilon_{\mathrm{GT}}(N,\delta)\big)_+. \tag{14}$$

Thus, when $u_{\mathrm{abst}} \geq 1/|\mathcal{Y}|$, the observable singleton rate $G_0$ directly bounds the best achievable selective utility. This is the abstention-side counterpart of the cliff: $M_0$ measures how much of the query distribution falls in the un-storable tail the cliff identifies, and $G_0$ estimates it from the sample alone, giving an observable handle on how much retrieval must cover. The bound is loosest for structured facts like PopQA and tightest in the pure arbitrary-facts regime, where it is a ceiling on what abstention and retrieval can fix in the tail.

Factual recall often involves near-misses (spelling variations) or partial structure, so we checked whether the sharp transition depends on strict binary (exact-match) scoring. We refit the sigmoid under five increasingly lenient regimes (strict binary, fuzzy match, token overlap, hand-crafted partial credit, Brier quality). The transition *shape* barely moved: neither the steepness ($\kappa \in [3.77, 4.93]$) nor the location ($\log_{10} f_{\mathrm{crit}} \in [4.09, 4.15]$) shifted systematically. What lenient scoring changed was the asymptotic error floor (0.158 to 0.430), as tail entities collected partial credit for uncertainty-aware outputs. Near-misses are only 5.6% of errors, so they do not drive the cliff: it sits where it sits regardless of how forgivingly we score, the signature of an internal capacity boundary rather than a scoring effect.

The framework's scope is delimited by two preconditions for the sigmoidal transition: (i) the task decomposes into discrete competency units drawn from a Zipfian distribution, and (ii) each unit has a *time-stable ground truth*. We verified the necessity of precondition (ii) by attempting to replicate the cliff in the code/API domain (using GitHub stars as an exposure proxy). The sigmoid entirely failed to emerge ($R^2 \approx 0$). Accuracy was non-monotonic because API defaults evolve across versions; models trained on multi-version documentation often answer confidently but incorrectly for the "current" default. Thus, while our framework naturally extends to domains like medical QA, legal citation, or translation (where facts are time-stable), it fundamentally does not apply to non-stationary environments or pure reasoning tasks (e.g., math proofs) where performance relies on logic depth rather than exposure coverage.

Evaluating the framework imposes a methodological requirement. Our results rely on PopQA, which is explicitly stratified by frequency. Generalizing to Mintaka (a standard, unstratified QA set), the global sigmoid degraded ($R^2 = 0.461$) and the pre-inference predictor underperformed raw confidence, because unstratified data severely undersamples the mid-frequency and tail regions and so denies the global fit its anchor points. The cause is structural, not a failure of the effect: within Mintaka relations that did have enough data, the sigmoid fits stayed excellent ($R^2 > 0.99$). The implication for future work is that applying the framework to a new domain means building a frequency-stratified evaluation set (the $\sim$75-sample bound of Section 5 sets the scale); standard benchmarks cannot simply be repurposed. We also checked that the cliff reflects genuine *entity-level exposure* rather than the particular choice of Wikipedia page views or some surface confound. Alternative structural proxies (e.g., alias count) predict correctness (AUROC = 0.684), while adversarial proxies (string length, alphabetical order) collapse to chance (AUROC $\approx$ 0.50).

To rule out the concern that frequency merely correlates with surface features (alias commonality, answer fluency, tokenization), we ran a nested-regression confound test. The standardized log-frequency coefficient is essentially unchanged as controls are added (+1.38 alone, +1.40 after adding alias count, answer length, and subject/answer GPT-2 token lengths), and a frequency-only predictor cross-validates far above a confounds-only one (AUROC = 0.817 vs. 0.665). Frequency survives the controls, so the cliff is an entity-exposure effect rather than a scoring or tokenization artifact.

A sharper question is whether page views track *actual training exposure*, unmeasurable for Mistral. We address this with OLMo-2-7B, whose Dolma corpus is fully open and queryable via infini-gram (Liu et al., 2024). The cliff replicates (page-view fit $\kappa = 4.15$, $R_w^2 = 0.990$, AUROC = 0.807), confirming the phenomenon transfers to an open model. Counterintuitively, even with true Dolma counts, raw corpus counts predict *worse* than page views (AUROC = 0.581 vs. 0.807). We attribute this to a distinction the theory makes but raw

counts ignore: storage is driven by *fact-encoding opportunities*, the structured co-mention of an entity with its relations, whereas raw *n*-gram counts measure surface-string frequency dominated by homonym collisions (the film "Cars" with the noun). Page-view prominence proxies encoding-opportunity more faithfully; consistent with this, on low-polysemy multi-word entities the page-view/raw-count correlation rises from $\rho = 0.41$ to $0.55$. We thus treat page views as a validated but imperfect exposure proxy, with direct co-occurrence measurement on an open corpus left to future work.

## 8   Conclusion

We have shown that factual reliability in LLMs exhibits a predictable, sigmoidal "knowledge cliff" driven by entity exposure: below a critical frequency $f_{\text{crit}}$ accuracy is poor, above it it rises rapidly to a plateau. The cliff is not a single global boundary; its location varies by more than an order of magnitude with the relational structure of the queried fact. Combining raw frequency with relational metadata gives a fully pre-inference risk score that captures most of the predictive signal and beats the model's own confidence in the rare-entity tail, supporting a two-stage routing architecture: pre-inference metadata for RAG triage, post-inference confidence held back for marginal refinement.

We also validate the prediction that the transition location is governed by model capacity. Across the Qwen2.5-Instruct family (fixed 18T-token corpus), $f_{\text{crit}}$ obeys a power law $f_{\text{crit}} \propto P^{-\alpha}$ with $\hat{\alpha} \approx 0.5$ (robust across model-exclusion and bootstrap perturbations), so a $10\times$ larger model reliably recalls entities roughly $3\times$ rarer. Comparing families suggests more curated corpora (e.g., Mistral) shift the intercept without changing the slope. But because tail accuracy stays near 10% regardless of model size, parameter scaling alone cannot close the long-tail deficit.

For tasks that decompose into Zipfian competency units, then, factual reliability is not a generation-time anomaly but a property of the query's exposure and structural difficulty, predictable in advance. Exposing it requires evaluation sets explicitly stratified by frequency; without that, global metrics mask the tail collapse. By formalizing the shape of the knowledge boundary, the framework moves hallucination mitigation from post-hoc output analysis toward pre-inference risk assessment and compute allocation.

**Code and data.**   Code for reproducing all experiments will be made available upon publication.

## Broader Impact Statement

Two considerations merit attention when deploying frequency-based risk scores. First, *allocative fairness*: because exposure proxies such as page views assign higher risk to entities from under-documented topics, regions, and communities, frequency-keyed routing or abstention could differentially decline to answer questions about long-tail, often marginalized, subjects. Deployments should monitor abstention rates across topical and demographic slices and prefer pairing frequency triage with retrieval (which our routing experiments show disproportionately helps low-frequency queries) over abstention alone. Second, *false assurance outside scope*: our scope-condition experiments show the framework fails silently under non-stationary ground truth (e.g., versioned APIs), where the score can look calibrated while being uninformative, so it should not be used outside time-stable factual QA without re-validation on a frequency-stratified set.

## A   Proofs

For readability we collect here the longer proofs deferred from the main text. Each is reproduced with its proposition statement number for reference.

### Proof of Proposition (Gradient saturation implies logarithmic encoding)

*Proof.* Write $g_m := \|\nabla_\theta f_k(\theta^{(m)})\|_2^2$. Since $\mathcal{L}'(u) = -\sigma(-u)$, the parameter step is $\Delta\theta_m = \eta\,\sigma(-u^{(m)})\,\nabla_\theta f_k(\theta^{(m)})$. By Taylor's theorem with a Lipschitz gradient, the margin update is $u^{(m+1)} - u^{(m)} = \eta\,\sigma(-u^{(m)})\,g_m + r_m$, with $|r_m| \leq C_r \eta^2 \sigma(-u^{(m)})^2 g_m$.

Define the transformation function $V(u) := u + e^u$, which has derivative $V'(u) = 1 + e^u = \frac{1}{\sigma(-u)}$. A second-order expansion gives:

$$V(u^{(m+1)}) - V(u^{(m)}) = V'(u^{(m)})(u^{(m+1)} - u^{(m)}) + \xi_m.$$

Substituting the margin update, the first-order term simplifies perfectly:

$$V'(u^{(m)})(u^{(m+1)} - u^{(m)}) = \frac{1}{\sigma(-u^{(m)})}\left[\eta\sigma(-u^{(m)})g_m + r_m\right] = \eta g_m + \frac{r_m}{\sigma(-u^{(m)})}.$$

Because $|r_m|$ scales with $\sigma(-u^{(m)})^2$, the remainder $\frac{r_m}{\sigma(-u^{(m)})}$ is bounded by $O(\eta^2)$. Similarly, the second-order term $\xi_m$ scales with $V''(u) = e^u$, which when multiplied by $(\Delta u)^2 \propto e^{-2u}$, decays exponentially and is strictly bounded by $O(\eta^2)$.

Summing over $M$ steps yields $V(u^{(M)}) = V(u^{(0)}) + \eta\sum_{m=1}^{M} g_m + O(M\eta^2)$. For large $u$, $V(u) \approx e^u$, which implies $e^{u^{(M)}} = \Theta(\eta M)$. Taking the logarithm yields $u^{(M)} = \log(\eta M) + O(1) + O(M\eta^2)$, proving (3). $\qquad\square$

**Proof of Proposition (Sigmoid phase transition)**

*Proof.* We proceed in two steps: we first solve the deterministic rate-distortion problem under threshold distortion; second, we then introduce storage heterogeneity. Assume all entities have identical storage cost ($\eta_k = 0$ for all $k$, i.e., no heterogeneity). Under Assumption 1, the threshold distortion is:

$$d_k(r_k) = \begin{cases} 0 & \text{if } r_k \geq r_{\min} \\ 1 & \text{if } r_k < r_{\min} \end{cases} \tag{15}$$

The rate-distortion problem $\min_{\{r_k\}} \sum_k q_k d_k(r_k)$ subject to $\sum_k r_k \leq R_{\text{total}}$ reduces to a 0-1 knapsack with equal weights: select the subset $S \subseteq \mathcal{K}$ maximizing $\sum_{k \in S} q_k$ subject to $|S| \cdot r_{\min} \leq R_{\text{total}}$.

Since all items have cost $r_{\min}$, the optimal solution stores the $K^* = \lfloor R_{\text{total}}/r_{\min}\rfloor$ entities with the highest $q_k$, i.e., entities with rank $k \leq K^*$. Under Zipf ordering ($q_1 \geq q_2 \geq \cdots$), this yields a step function:

$$\text{acc}_{\text{det}}(f) = \begin{cases} 1 & \text{if } f \geq f_{K^*} \\ 0 & \text{if } f < f_{K^*} \end{cases} \tag{16}$$

where $f_{K^*}$ is the frequency of the marginal stored entity.

This deterministic, homogeneous case fixes the *location* of the boundary: the cutoff sits at the frequency $f_{K^*}$ of the marginal stored entity, set by the capacity budget. What it does not capture is the *shape* of the transition, which is hard only because every entity is assumed equally difficult to store. Introducing heterogeneity in storage difficulty smooths the step into a graded transition; we now incorporate Assumptions 2 and 3. Entity $k$ requires rate $r_{\min,k} = r_0 + \eta_k/\kappa'$ (Assumption 2), and training provides effective rate $r_k^{\text{eff}} = \beta \log f_k$ (Assumption 3). Entity $k$ is successfully stored if $r_k^{\text{eff}} \geq r_{\min,k}$, i.e., if $\beta \log f_k \geq r_0 + \eta_k/\kappa'$, which rearranges to:

$$\eta_k \leq \kappa(\log f_k - \log f_{\text{crit}}) \tag{17}$$

where $\kappa = \kappa'\beta$ absorbs the proportionality constants and $\log f_{\text{crit}} = r_0/\beta$ is the critical log-frequency at which the median entity is stored. Treating $\eta_k$ as drawn from a standard logistic distribution across entities (Assumption 2), the probability that an entity of frequency $f_k$ is stored is:

$$P(\text{stored} \mid f_k) = P\bigl(\eta_k \leq \kappa(\log f_k - \log f_{\text{crit}})\bigr) = \sigma\bigl(\kappa(\log f_k - \log f_{\text{crit}})\bigr) \tag{18}$$

where $\sigma(x) = 1/(1 + e^{-x})$ is the logistic sigmoid. Let $a_{\max}$ denote expected accuracy when the fact is successfully stored and $a_{\min}$ expected accuracy when it is not. Then

$$\text{acc}(f) = a_{\max}P(\text{stored} \mid f) + a_{\min}\bigl(1 - P(\text{stored} \mid f)\bigr),$$

which yields Equation (4) after substituting Equation (18). As a consistency check, the homogeneous limit is recovered: as heterogeneity vanishes ($\kappa \to \infty$), $\sigma(\kappa(\log f - \log f_{\text{crit}}))$ approaches the indicator $\mathbf{1}[f \geq f_{\text{crit}}]$, returning the hard step function of Equation (16). $\qquad\square$

**Proof of Proposition (Data-determined cliff)**

*Proof.* Consider entities ranked $k = 1, 2, \ldots, K$ by descending frequency. Under the Zipfian distribution with exponent $\alpha$, the probability of the $k$-th entity is:

$$p_k = \frac{k^{-\alpha}}{H_K^{(\alpha)}}, \quad \text{where } H_K^{(\alpha)} = \sum_{j=1}^{K} j^{-\alpha} \tag{19}$$

is the generalized harmonic number. For a corpus of $N$ total training tokens, the expected frequency is $f_k = N \cdot p_k = (N/H_K^{(\alpha)}) \cdot k^{-\alpha}$. The critical frequency of Proposition 4 is the median-storage point ($\eta_k = 0$, where the effective rate meets the baseline requirement $r_0$); because the heterogeneity is centered about the homogeneous threshold, this coincides with the marginal stored entity of the knapsack, at rank $K^* = \lfloor R_{\text{total}}/r_{\min} \rfloor$. Substituting into the frequency model:

$$f_{\text{crit}} = f_{K^*} = \frac{N}{H_K^{(\alpha)}} \cdot (K^*)^{-\alpha} \approx \frac{N}{H_K^{(\alpha)}} \cdot \left(\frac{R_{\text{total}}}{r_{\min}}\right)^{-\alpha} \tag{20}$$

Taking logarithms: $\log f_{\text{crit}} = \log N - \log H_K^{(\alpha)} - \alpha \log R_{\text{total}} + \alpha \log r_{\min}$. To express this in terms of corpus size alone, we apply Heaps' law: $K \approx \eta N^\beta$ for $\beta \in [0.4, 0.6]$. The scaling of $H_K^{(\alpha)}$ depends on $\alpha$. For $\alpha \approx 1$, $H_K^{(1)} \approx \log K + \gamma$, by Euler–Mascheroni, so $\log H_K \approx \log(\beta \log N + \log \eta + \gamma)$. Because $\log(\log N)$ grows sub-polynomially, this term becomes negligible relative to $\log N$ for large corpora. Conversely, for $\alpha < 1$, $H_K^{(\alpha)} \approx K^{1-\alpha}/(1-\alpha)$, giving $\log H_K^{(\alpha)} \approx (1-\alpha)\beta \log N + c_0$. Unifying these regimes, we obtain: $\log f_{\text{crit}} = c_{\text{corpus}} \log N - \alpha \log R_{\text{total}} + C(\alpha, r_{\min})$ where $c_{\text{corpus}} = 1$ for $\alpha \approx 1$, and $c_{\text{corpus}} = 1 - (1-\alpha)\beta$ for $\alpha < 1$. The term $C(\alpha, r_{\min})$ is a constant independent of $N$ and $R_{\text{total}}$. □

**Proof of Proposition (Bayes-optimal pre-inference AUROC and converse bound)**

*Proof.* For part (i): $\text{AUROC}(s) = P(s(X_+) > s(X_-))$ where $X_+$ and $X_-$ are drawn from $p(x \mid y = 1) \propto g(x)q(x)$ and $p(x \mid y = 0) \propto (1 - g(x))q(x)$ respectively. By the Neyman–Pearson lemma, the likelihood ratio $g(x)/(1 - g(x))$ is the optimal test statistic for separating $X_+$ from $X_-$. Since $g(x)$ is strictly increasing in $x$ and $h(t) = t/(1 - t)$ is strictly increasing on $(0, 1)$, the composite $g(x)/(1 - g(x))$ is strictly increasing in $x$. Since AUROC depends only on the ranking induced by $s$ (not its values), any strictly monotone transformation of an optimal statistic is also optimal. Because $x \mapsto g(x)/(1 - g(x))$ is strictly increasing, $x$ induces the same ranking as the likelihood ratio, so the frequency score $s^*(x) = x$ is an equivalent optimal statistic. For the closed form, write $\text{AUROC}^* = P(X_+ > X_-)$ and substitute the normalized class-conditional densities $p(x \mid y = 1) = g(x)q(x)/\bar{a}$ and $p(t \mid y = 0) = (1 - g(t))q(t)/(1 - \bar{a})$; by the Wilcoxon–Mann–Whitney identity,

$$\begin{aligned}
\bar{a}(1 - \bar{a}) \cdot \text{AUROC}^* &= \iint_{x>t} g(x)(1 - g(t)) \, q(x)q(t) \, dx \, dt, \\
&= \iint_{x>t} g(x) \, q(x)q(t) \, dx \, dt - \iint_{x>t} g(x)g(t) \, q(x)q(t) \, dx \, dt. \tag{21}
\end{aligned}$$

Integrating out $t$ first gives $\int g(x) \, F_q(x) \, q(x) \, dx = \mathbb{E}_q[g(X)F_q(X)]$. The integrand $g(x)g(t)q(x)q(t)$ is symmetric in $(x, t)$, so the integral over the half-plane $\{x > t\}$ equals exactly half the integral over the full plane, giving $B = \bar{a}^2/2$. Therefore:

$$\bar{a}(1 - \bar{a}) \cdot \text{AUROC}^* = \mathbb{E}_q[g(X)F_q(X)] - \frac{\bar{a}^2}{2}.$$

Since $\mathbb{E}_q[F_q(X)] = \frac{1}{2}$ (the expected value of a CDF under its own distribution), we have $\mathbb{E}_q[g(X)F_q(X)] = \text{Cov}_q(g(X), F_q(X)) + \bar{a} \cdot \frac{1}{2}$. Substituting:

$$\bar{a}(1 - \bar{a}) \cdot \text{AUROC}^* = \text{Cov}_q(g(X), F_q(X)) + \frac{\bar{a}}{2} - \frac{\bar{a}^2}{2} = \text{Cov}_q(g(X), F_q(X)) + \frac{\bar{a}(1 - \bar{a})}{2},$$

yielding (9) after dividing by $\bar{a}(1 - \bar{a})$. For part (ii): the converse is immediate from Neyman–Pearson optimality of the likelihood ratio. For strict monotonicity in $\kappa$, we use a first-order stochastic dominance (FOSD) argument. Fix $\kappa_2 > \kappa_1 > 0$. Since $\sigma(\kappa u)$ is increasing in $\kappa$ for $u > 0$ and decreasing for $u < 0$, the accuracy functions satisfy

$$g_{\kappa_2}(x) > g_{\kappa_1}(x) \ \text{ for } x > x_{\text{crit}}, \qquad g_{\kappa_2}(x) < g_{\kappa_1}(x) \ \text{ for } x < x_{\text{crit}},$$

so $g_{\kappa_2}(x)/g_{\kappa_1}(x)$ crosses 1 from below exactly once at $x_{\text{crit}}$. Consider the positive-class densities $p_{+,\kappa}(x) \propto g_\kappa(x)\, q(x)$ with normalizing constant $\bar{a}_\kappa$. Their ratio is

$$\frac{p_{+,\kappa_2}(x)}{p_{+,\kappa_1}(x)} = \frac{g_{\kappa_2}(x)}{g_{\kappa_1}(x)} \cdot \frac{\bar{a}_{\kappa_1}}{\bar{a}_{\kappa_2}}.$$

Since $\bar{a}_{\kappa_1}/\bar{a}_{\kappa_2}$ is a positive constant, multiplying by it preserves the single-crossing property: the ratio $p_{+,\kappa_2}/p_{+,\kappa_1}$ still crosses 1 exactly once from below (at a point near $x_{\text{crit}}$). By the standard single-crossing criterion for FOSD, $p_{+,\kappa_2}$ FOSD-dominates $p_{+,\kappa_1}$. An identical argument with $1 - g_\kappa$ shows $p_{-,\kappa_1}$ FOSD-dominates $p_{-,\kappa_2}$ (the ratio $(1 - g_{\kappa_2})/(1 - g_{\kappa_1})$ crosses 1 from above exactly once at $x_{\text{crit}}$). These two shifts compound rather than cancel: $P(X_+ > X_-)$ is increasing in $X_+$ and decreasing in $X_-$ (stochastically), and the steeper cliff shifts $X_+$ rightward and $X_-$ leftward, so both moves raise $P(X_+ > X_-)$. Therefore

$$\text{AUROC}^*(\kappa_2) = P(X_{+,\kappa_2} > X_{-,\kappa_2}) \geq P(X_{+,\kappa_1} > X_{-,\kappa_1}) = \text{AUROC}^*(\kappa_1).$$

Strictness follows because $\Pr_q(X < x_{\text{crit}}) \in (0, 1)$ (the cliff lies strictly inside the support of $q$), so the FOSD inequalities are strict on sets of positive $q$-measure.

The limits follow by $g_\kappa \to \bar{a}$ uniformly as $\kappa \to 0$ (so $\text{AUROC}^* \to \frac{1}{2}$) and $g_\kappa \to a_{\min} + (a_{\max} - a_{\min})\, \mathbf{1}[x > x_{\text{crit}}]$ as $\kappa \to \infty$. For the infinite-steepness limit, $g(x)$ becomes a step function. Let $P_< = \Pr_q(X < x_{\text{crit}})$ and $P_> = \Pr_q(X > x_{\text{crit}})$. Evaluating the covariance integral directly yields:

$$\mathbb{E}_q[g(X)F_q(X)] = \int_{x < x_{\text{crit}}} a_{\min} F_q(x) q(x) dx + \int_{x > x_{\text{crit}}} a_{\max} F_q(x) q(x) dx$$
$$= \frac{1}{2} a_{\min} P_<^2 + \frac{1}{2} a_{\max}(1 - P_<^2).$$

Subtracting $\bar{a}/2 = \frac{1}{2}(a_{\min} P_< + a_{\max} P_>)$ gives $\text{Cov}_q(g, F_q) = \frac{1}{2} P_< P_> (a_{\max} - a_{\min})$. Plugging this exact covariance into (9) immediately yields the stated upper bound limit.

$$\square$$

**Proof of Proposition (Sample complexity of cliff localization)**

*Proof.* For part (i), the log-likelihood of a single observation is $\ell(x, y) = y \log \sigma(u) + (1 - y) \log(1 - \sigma(u))$ where $u = \kappa(x - x_{\text{crit}})$. Using the identity $\sigma'(u) = \sigma(u)(1 - \sigma(u))$, the score function is $\partial\ell/\partial x_{\text{crit}} = -\kappa \frac{\partial\ell}{\partial u} = \kappa(\sigma(u) - y)$. Since $\mathbb{E}[y] = \sigma(u)$, the Fisher information is $\mathbb{E}[(\partial\ell/\partial x_{\text{crit}})^2] = \kappa^2 \text{Var}(y) = \kappa^2 \sigma(u)(1 - \sigma(u)) = \kappa^2 \sigma'(u)$, confirming part (i). For part (ii), integrate against $p(x) = 1/X$:

$$\mathcal{I}_n(x_{\text{crit}}) = \frac{n\kappa^2}{X} \int_0^X \sigma'(\kappa(x - x_{\text{crit}}))\, dx = \frac{n\kappa}{X} \int_{-\kappa x_{\text{crit}}}^{\kappa(X - x_{\text{crit}})} \sigma'(u)\, du.$$

For $x_{\text{crit}}$ sufficiently far from the boundaries ($\kappa x_{\text{crit}} \gg 1$ and $\kappa(X - x_{\text{crit}}) \gg 1$), the integral of the PDF $\sigma'(u)$ approaches 1, yielding $\mathcal{I}_n \approx n\kappa/X$. The sample bound follows from the Cramér–Rao bound $\text{Var}(\hat{x}_{\text{crit}}) \geq \mathcal{I}_n^{-1}$ and asymptotic normality.

For part (iii), integrate against $p(x) = 10^{(1-\alpha)x}/Z_\alpha$. Under the substitution $u = \kappa(x - x_{\text{crit}})$, we expand $10^{(1-\alpha)x} = 10^{(1-\alpha)(x_{\text{crit}} + u/\kappa)} = f_{\text{crit}}^{1-\alpha} e^{tu}$, where $t = (1 - \alpha)\ln(10)/\kappa$. Extending the integration to $\mathbb{R}$ gives:

$$\mathcal{I}_n(x_{\text{crit}}) \approx \frac{n\kappa \cdot f_{\text{crit}}^{1-\alpha}}{Z_\alpha} \int_{-\infty}^\infty \sigma'(u)\, e^{tu}\, du.$$

The integral is exactly the moment-generating function of the standard logistic distribution, which evaluates to $\pi t/\sin(\pi t)$ for $|t| < 1$. Continuity as $t \to 0$ ($\alpha \to 1$) smoothly recovers part (ii). $\square$

# B  Supplementary Analysis

### Proof of the missing-mass utility bound

*Proof.* Fix the sample $S$. Decompose by whether the test prompt is seen in $S$. On seen prompts, utility is at most 1. For an unseen prompt in the arbitrary-facts model, conditional on the training sample (and any learner state derived from it), the true label remains uniform on $\mathcal{Y}$ and independent of the predictor's output. Hence any non-abstaining prediction has expected utility at most $1/|\mathcal{Y}|$, whereas abstention yields utility $u_{\text{abst}}$. Therefore, when $u_{\text{abst}} \geq 1/|\mathcal{Y}|$, abstention weakly dominates guessing (and strictly dominates it when $u_{\text{abst}} > 1/|\mathcal{Y}|$). Thus, $\mathbb{E}[u \mid S] \leq (1 - M_0(S)) \cdot 1 + M_0(S) \cdot u_{\text{abst}} = 1 - (1 - u_{\text{abst}})M_0(S)$. $\qquad\square$

### Robustness of the logistic form

The logistic approximation $\hat{a}(f)$ is robust to the misspecification of the true storage heterogeneity distribution. Let $F_{r^*}$ denote the true cumulative distribution function (CDF) of the latent storage thresholds across entities. We define the approximation error as $\delta(x) = F_{r^*}(x) - \sigma(x)$, where $x = \kappa(\log f - \log f_{\text{crit}})$ is the standardized log-frequency. Since $\hat{a}(f) - a(f) = (a_{\max} - a_{\min})\delta(x)$, writing $\mathcal{E}$ as an expectation and applying $|\delta| \leq d_{\text{KS}} := \sup_x |\delta(x)|$ gives $\mathcal{E} := \mathbb{E}_q\big[(\hat{a}(f) - a(f))^2\big] \leq (a_{\max} - a_{\min})^2 \cdot d_{\text{KS}} \cdot \mathbb{E}_q[|\delta(X)|]$, where $\mathbb{E}_q[|\delta(X)|]$ is the average pointwise deviation under the query distribution. The bound has a natural interpretation: calibration error is controlled by the amplitude squared, the worst-case pointwise error ($d_{\text{KS}}$), and the average pointwise error ($\mathbb{E}_q[|\delta|]$). Let $L_f$ be the effective width of the log-frequency support. At $\alpha = 1$, $\mathbb{E}_q[|\delta|] \approx W_1(F_{r^*}, \text{Logistic})/(\kappa L_f)$ by the CDF identity $\int |F - G|\, dx = W_1(F, G)$; for $\alpha > 1$ the expectation is smaller since heavier Zipf concentrates queries on low-$x$ entities where $|\delta| \approx 0$.

A steep cliff is self-correcting: the transition window has width $O(1/\kappa)$ in $\log f$, outside which $|\delta(x)| \approx 0$ for any unimodal $F_{r^*}$, so misspecifying the distribution family costs negligible calibration error. For Mistral-7B at $\alpha = 1$, with $a_{\max} - a_{\min} = 0.650$, $d_{\text{KS}} = 0.203$, and $\mathbb{E}_q[|\delta|] \approx W_1/(\kappa L_f) = 0.5/34.09 \approx 0.015$, the bound gives $\mathcal{E} \lesssim 0.0013$, consistent with the observed $R_w^2 = 0.969$.

By the DKW inequality (Massart, 1990), replacing $d_{\text{KS}}$ with its empirical estimate $\hat{d} := \sqrt{\log(2/\nu)/(2n)}$ and bounding $\mathbb{E}_q[|\delta|]$ via $W_1/(\kappa L_f)$, $\mathcal{E} < \tau$ can be certified with probability $\geq 1 - \nu$ whenever

$$n \geq \frac{(a_{\max} - a_{\min})^4 \cdot W_1^2 \cdot \log(2/\nu)}{2\,(\kappa L_f)^2\, \tau^2}.$$

The $1/\kappa^2$ scaling confirms that steep cliffs are quadratically cheaper to certify, linking to Proposition 8.

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
