# OpenReview forum: "Predicting LLM Hallucination Risk from Entity Frequency: A Rate-Distortion Perspective"
_TMLR — Under review for TMLR_

### Review · Reviewer_TXhz · 2026-05-06

**Summary Of Contributions:**

The paper studies LLMs ability to recall facts from their training data using the perspective of rate-distortion theory. The paper first establishes that recall quality is linked to the frequency of a particular fact in the training data: the higher the frequency, the lower the probability of hallucination. In particular, the paper finds that there is a sharp threshold below which recall is at the worst possible value. This cliff turns to a smooth, sigmoidal transition when facts have different recall difficulties. The paper then derives a number of results including that information allocation per fact is proportional to its log frequency, and that fact frequency is, in a sense, the optimal discriminator for hallucination risk. Experiments on LLMs then demonstrate that pretraining capacity fundamentally determines hallucination risk (in the sense that fine-tuning cannot improve the critical frequency to recall facts), that frequency can indeed predict hallucination risk without any inference, and that model capacity reduces hallucination risk.

**Audience:**

Yes

**Audience Explanation:**

Yes: predicting hallucination risk in LLMs is an important and timely problem, and this paper proposes a pre-inference method to predict this risk. Moreover, the paper produces a number of theoretical results which may be useful to others in related fields outside of purely LLMs. Overall, a paper with a likely strong impact in the field of hallucination detection and the theory of factuality and reliability of LLMs.

**Broader Impact Concerns:**

No broader impact concerns.

**Claims And Evidence:**

Yes

**Claims Explanation:**

The theoretical results in the paper are solid: proofs appear correct and assumptions are well supported. In fact, the theory is quite extensive. The remarks made throughout the paper address common questions or objections that a reader may have.

Although experiments are not the main focus of this paper, they are also well-conducted: important ablations are included and theoretical assumptions are validated. Indeed, even the experiments are fairly extensive: the authors produce 6 different experiments in addition to controls and other validations.

My primary concern with the paper is that its length detracts from its clarity. Nearly all 29 pages of the paper are quite dense with theory and experimental results.  I understand the need to keep all key results in the main text, but I would suggest that the authors consider putting some proofs or secondary results (both theoretical and empirical) in the appendix. This can help the reader focus on the key takeaways from this work and not get lost in the details.

**Requested Changes:**

**Would strengthen**
- Text in Figures 3 and beyond is too small; please increase font size
- Consider moving some results to the appendix

---

> ### Author Response · Authors · 2026-06-17
> **Response to Reviewer TXhz**
>
> We thank the reviewer for the positive assessment. A complete response to all reviewers is in the **Response to Reviewers PDF** (supplementary material).
>
> We thank the reviewer for the positive assessment and helpful suggestions.
>
> ### 1. Figure fonts
>
> All figures from Figure 3 onward have been regenerated with larger font sizes.
>
> ### 2. Move results to appendix
>
> We have moved the five longest proofs (Propositions 3, 4, 5, 9, and 11) to a new Deferred Proofs appendix, leaving proof sketches with pointers in the main text, and additionally moved a dense secondary robustness analysis (the logistic-form robustness remark) to a Supplementary Analysis appendix. We also tightened the main text throughout. The paper is now 29 pages (from 31), with the main text reduced to 26.5 pages, even with the new experiments added, so we hope this directly addresses the density concern while keeping the main results and their short proofs inline.

---

### Review · Reviewer_ctJY · 2026-05-24

**Summary Of Contributions:**

This paper studies whether the hallucination risk of an LLM can be predicted before running the model, using the frequency or popularity of the entity in the query. The paper proposes a rate-distortion-based theory to explain this phenomenon and derives a “knowledge cliff,” where accuracy changes quickly around a critical frequency. It also studies how relation type, model size, and retrieval routing affect this prediction problem.

**Audience:**

No

**Audience Explanation:**

The paper may have limited appeal to the target audience in its current form. The topic itself is potentially interesting, since pre-inference hallucination risk prediction could be useful for retrieval-augmented systems. However, the current manuscript does not present this idea in a way that clearly serves either an empirical or a theoretical audience. Readers interested in practical hallucination detection may find the experimental evidence too limited and the proposed method not sufficiently convincing beyond the known observation that popular entities are easier for LLMs to answer correctly. Readers interested in theory may find the rate-distortion model too artificial, with many key assumptions and concepts insufficiently defined or justified.

**Claims And Evidence:**

No

**Claims Explanation:**

1.The contribution part in the introduction contains so many complicated concepts, which have not been introduced when readers reach that part. It should focus more on general knowledge.

2.The studied setting is very artificial. For example, “An LLM $M$ is trained on a corpus of entities $K = {1, . . . , K}$, where $f_k \propto k^{−α}$ is the Zipf frequency of the $k$-th most common entity”, why the finite entity assumption? Why is the Zipf frequency assumption? Neither of these has been strictly verified. Then, why do you consider “bits” of knowledge? Why is the hallucination probability $d_k(r_k)$ defined to depend on bits? I would not say it does not make any sense, as the authors cited prior work. However, the current argument is not convincing, leaving me doubtful about the fundamental capacity allocation problem it tries to study.

3.Why can the distortion function be assumed to be differentiable, strictly convex, and strictly decreasing? More importantly, what is a distortion function?

4.Proposition 3: “$f_k$ denotes the network’s pre-softmax margin for the correct answer” What does this mean?Which network? What is pre-softmax? What is margin for the correct answer? Is it the same as the entity distribution induced by the corpus?$u^{(M)} = \log(\eta M) + O(1) + O(M \eta^2)$ What is the meaning for this bound?

5.The derivation of the sigmoid “knowledge cliff” needs clearer justification. The paper first presents a convex rate-distortion allocation problem with a differentiable, strictly convex, and strictly decreasing distortion function, and later derives the sigmoid curve from threshold distortion, heterogeneous storage difficulty, and logarithmic rate allocation. It is unclear which part of the sigmoid behavior is a consequence of the rate-distortion formulation itself, and which part comes from the additional assumptions on threshold storage and the distribution of storage difficulty.

6.The connection between the theoretical quantities and the empirical measurements is unclear. The theory uses the true pretraining frequency of an entity, the allocated number of bits, and the effective storage rate. However, the experiments appear to use external popularity proxies, such as Wikipedia statistics or pageviews, rather than the actual occurrence frequency in the model’s pretraining corpus. This is a significant gap because the main theoretical claim is about training exposure and capacity allocation, while the empirical evidence may only show that public popularity correlates with factual accuracy.

7.Many introduced concepts are complicated but useless: “Mandelbrot-Zipf” even Zipf is not well clarified. “calibration” not defined at all. “The Erasure Regime” what’s this? “Operational Routing”, “Rate–Distortion Geometry” not well defined. “AUROC” not defined … I do not list them all. I believe listing these issues can already prove that the writing is terrible and loses the rigor that an academic paper requires.

**Requested Changes:**

I believe the manuscript requires a comprehensive rewrite. In particular, the authors should clearly define all introduced concepts before using them, reduce the density of technical terminology in the introduction, and reorganize the results into a smoother and more coherent narrative. Moreover, the authors should reconsider whether the overall story is well justified and whether the experiments provide sufficient support for the paper’s main claims.

---

> ### Author Response · Authors · 2026-06-17
> **Response to Reviewer ctJY**
>
> We thank the reviewer for pushing us on clarity and rigor. We have substantially revised the manuscript; a complete point-by-point response is in the **Response to Reviewers PDF** (supplementary material). One clarification underlies several points: we adapt the rate--distortion *framework* to factual recall rather than applying it literally; "distortion" here is the probability of hallucination, and the distortion-rate curve is posited and validated empirically, not derived from a source code. We now state this explicitly. In brief:
>
> **1. Intro too dense.** The Contributions are rewritten so each item opens with a plain-language sentence before any technical term, preceded by a one-sentence non-technical summary. Technical concepts ($\kappa$, AUROC\*, Fisher information) now appear only after the intuition, and all are defined in a new Preliminaries (Section 3.1).
>
> **2. Artificial setting (Zipf, finite $K$, bits).** A new paragraph addresses each idealization: the Zipfian form is inherited from the corpus, not assumed (and verified robust via Mandelbrot--Zipf); finite $K$ is an indexing device, not a claim about world facts; the capacity figure only establishes that total rate is finite and sub-lossless. The load-bearing assumption (logarithmic rate) is directly validated: accuracy fits far better on $\log f$ than raw $f$ ($R^2=0.958$ vs $0.817$), and the heterogeneity assumption is confirmed by the order-of-magnitude spread in relation-specific $f_{\text{crit}}$.
>
> **3. Distortion function: definition and assumptions.** The new Preliminaries defines it in plain language: it maps bits of allocated capacity to the probability the model fails to recall a fact. *Decreasing* is definitional (more capacity, fewer errors); *convexity* encodes diminishing returns (each additional bit helps less than the last) and is the standard shape of a rate--distortion curve. We clarify that convexity is not a primitive assumption: our per-fact distortions are threshold (step) functions; the smooth convex curve arises as the population average over heterogeneous thresholds, with the sigmoid as the logistic-mixture special case.
>
> **4. Prop 3 margin unclear.** We define the margin where Proposition 3 is stated: the logit (pre-softmax score) of the correct answer minus that of the strongest competitor; a positive margin means the correct answer is most probable. Supporting terms are in Preliminaries, and the margin-as-rate link is now an explicit modeling assumption.
>
> **5. Which part of the sigmoid comes from which assumption.** A new roadmap paragraph separates the ingredients: the *existence* of a cliff follows from capacity-constrained allocation over a Zipfian prior (Proposition 2); the *sharpness* from threshold distortion; the smooth *sigmoidal shape* from heterogeneous storage difficulty; and the *placement* on the log-frequency axis from logarithmic rate allocation.
>
> **6. Theory-experiment gap (page views vs true frequency).** A fair point, which we answer with a new experiment. The primary evidence that the effect is entity-level exposure (not surface popularity) is a confound test: the frequency signal is unchanged controlling for alias count, answer length, and tokenization (standardized coefficient near $+1.4$), and frequency alone (AUROC $0.817$) far exceeds those confounds combined ($0.665$). We also ran an open-corpus evaluation on **OLMo-2-7B** (Dolma): the cliff replicates (AUROC $0.807$), and raw corpus counts predict *worse* than page views ($0.581$), which we read as a statement about proxies (raw counts conflate homonyms) rather than a failure of the exposure hypothesis; controlling for polysemy supports this (the page-view/count correlation rises $0.41\to0.55$ on unambiguous multi-word entities). We now state directly that popularity is what we measure, exposure is what we theorize, and the link is supported but imperfect. We also cite concurrent interpretability work (Merullo et al., ICLR 2025) independently finding a co-occurrence-frequency threshold for reliable fact encoding.
>
> **7. Undefined terms.** We made a pass to simplify the writing: removed coined terms that added little (the "erasure regime" terminology is replaced by plain "low-frequency regime"), renamed two over-technical section titles, and defined Zipf/Mandelbrot--Zipf, calibration, ECE, and AUROC in the new Preliminaries so the paper is self-contained.
>
> **8. Comprehensive rewrite.** We restructured for readability: the new Preliminaries (3.1), plain-language Contributions, a roadmap paragraph (3.2), the five longest proofs moved to a Deferred Proofs appendix (sketches with pointers left inline), and explicit justification of the modeling assumptions. The main text is now 26.5 pages (the paper 29, from 31) even with the new experiments added. We are glad to make further structural changes if specific sections remain unclear.

---

### Review · Reviewer_jahR · 2026-06-09

**Summary Of Contributions:**

The paper proposes a query-dependent rate–distortion (RD) account of factual reliability in LLMs. The central object is a "knowledge cliff": modeling factual recall as lossy compression under a finite per-parameter capacity budget allocated over a Zipfian distribution of entities, the authors derive that accuracy is an affine sigmoid in log-frequency, with a critical frequency $f_{\text{crit}}$ marking the transition from an "erasure" regime (zero allocated rate) to a "stored" regime. From this they develop five results:

(1) a sigmoid phase-transition derivation under threshold distortion + heterogeneous storage difficulty + logarithmic rate allocation, with a gradient-saturation argument for the logarithmic term;

(2) a zero-compute, pre-inference frequency risk score plus a Bayes-optimal AUROC converse bound for any frequency-only predictor;

(3) relation-conditioned cliffs (a reported 76× range in $f_{\text{crit}}$) that supply within-stratum/individual-level discrimination and a retrieval-routing policy with an LP-duality "upgrade-by-gain" structure;

(4) a capacity scaling law $f_{\text{crit}} \propto P^{-0.52}$ on the Qwen2.5 family;

and (5) a Fisher-information sample-complexity bound for cliff localization. Experiments use 500 stratified PopQA queries on Mistral-7B/Falcon-7B/Qwen-2.5-7B (with Wikipedia page views as the exposure proxy), plus a QLoRA fine-tuning study, cross-model transfer, relation-specific fits, and the 5-point Qwen scaling sweep.

**Strengths.**

The framework is coherent and ties a well-documented empirical regularity (long-tail factual degradation) to a mechanistic explanation that yields falsifiable, decision-relevant predictions rather than a post-hoc fit. The connection to Kalai & Vempala / Kalai et al. (binary scoring = threshold distortion as the source of the cliff's *sharpness*) is genuinely clarifying, and the scope-conditions section (failure on code/API due to non-stationarity; failure on unstratified Mintaka) is unusually candid and strengthens credibility. The negative controls (shuffled frequency, adversarial proxies) are appropriate.

**Weaknesses.**

Several headline claims are stated more strongly than the evidence supports: the scaling-law exponent rests on $n=5$ with one excluded degenerate fit; the "beats confidence" comparison uses only mean-token-probability rather than the stronger uncertainty estimators the paper itself cites; the "99.2% of ceiling" framing is partly circular (ceiling and score both derive from the same fitted $\kappa$); and the fine-tuning conclusion is over-generalized from one narrow intervention. There is also at least one internal contradiction in the converse-bound numbers (see Requested Changes #1) and multiple inconsistent $f_{\text{crit}}$ values across tables/figures.

**Audience:**

Yes

**Audience Explanation:**

Pre-inference, zero-compute hallucination triage and RAG routing are of clear practical interest to the LLM-reliability and retrieval communities, and the RD framing that links the sharpness of the cliff to binary scoring (connecting Kalai et al. to a deployable predictor) is a conceptual contribution that practitioners and theorists alike will want to engage with. Even readers skeptical of the precise scaling exponent will find value in the scope conditions (when the cliff does/doesn't appear), the demonstration that fine-tuning does not shift the cliff left, and the within-stratum result that relation structure — not frequency — carries the discriminative signal in the rare-entity tail. The audience-interest bar is comfortably met.

**Broader Impact Concerns:**

No major ethical red flags; a brief Broader Impact Statement would suffice. Two points worth addressing: (i) fairness of the exposure proxy — page-view/frequency-based risk scoring may systematically flag entities from under-documented topics or communities as "unreliable," and routing/abstention policies keyed on frequency could differentially deny answers about long-tail (often marginalized) subjects; the authors should acknowledge this allocative dimension. (ii) false assurance / misapplication — the scope conditions show the framework fails silently on non-stationary or reasoning domains (the code/API $R^2\approx0$ case), where a deployed pre-inference score could appear calibrated while being meaningless; a sentence cautioning against transferring the score outside time-stable factual QA would be appropriate.

**Claims And Evidence:**

No

**Claims Explanation:**

The central empirical claim — that accuracy follows a sigmoidal cliff in log-frequency on frequency-stratified factual QA — is convincingly supported ($R^2_w > 0.89$ across three models, sensible negative controls, robustness to graded-correctness re-scoring). My "No" is driven by a small number of secondary-but-headlined claims whose evidence does not yet match their phrasing:

1. Converse-bound numbers contain an internal contradiction. Proposition 9 proves AUROC* is strictly increasing in $\kappa$. Yet Remark 7 reports Mistral ($\kappa = 4.87$) giving $AUROC* = 0.816$ and Falcon ($\kappa = 2.10$) giving $AUROC* = 0.837$, i.e. the shallower cliff yields a higher ceiling, then describes it as "the harder discrimination problem." This directly violates the monotonicity the paper just proved (unless $q$ differs across models, which is not stated). Until reconciled, the quantitative converse-bound results are not trustworthy.

2. The "99.2% of the theoretical upper bound" claim is partly circular. Both $AUROC*$ (via Eq. 24) and the empirical score derive from the same fitted sigmoid / $\kappa$ on the same data. Since $s*(x) = g(x)$ is optimal by construction, near-saturation of the ceiling is close to tautological and does not independently validate the predictor. This should be reframed as a consistency check, with the ceiling computed on held-out $\kappa$.

3. Frequency "outperforms the LLM's own confidence" rests on the weakest confidence baseline. The only inference-time baseline actually compared is mean token probability (AUROC 0.772). The paper cites semantic entropy (Kuhn et al.), $P(\text{True})$ (Kadavath et al.), and SelfCheckGPT but does not benchmark against them. The "frequency beats confidence" claim (abstract, conclusion) is central and currently under-supported.

4. The scaling law $f_{\text{crit}} \propto P^{-0.52}$ is fit on $n = 5$, with Qwen2.5-7B excluded from the visualized curves for a "degenerate sigmoid fit" yet retained in the regression. With one near-bound $\kappa$ and 5 points, the reported $\pm 0.05$ on the exponent is almost certainly optimistic. The cross-family intercept argument (Mistral 0.19 log-decades below the line) is plausible but is a one-point post-hoc observation.

5. Inconsistent $f_{\text{crit}}$ values undermine clarity. Mistral appears as 12,726 (Table 1) and 13,734 (Fig. 3 / Sec. 5); Qwen-7B as 50,843 (Table 1) vs 24,444 (Sec. 6.8, attributed to ChatML vs generic template); Table 3 base 24,802 (different split). The template sensitivity in particular (a roughly 2x shift in a quantity the whole paper treats as a stable model "fingerprint") deserves explicit discussion, since it suggests $f_{\text{crit}}$ is also a function of prompting protocol.

Additionally, the proxy itself (Wikipedia page views as a stand-in for training exposure) is never directly validated against measured pretraining counts; the triangulation is reassuring but indirect, and the framework's "exposure" interpretation rests entirely on it.

**Requested Changes:**

1. Resolve the Falcon vs Mistral $\text{AUROC}^*$ contradiction (Remark 7) against the monotonicity-in-$\kappa$ result (Prop. 9). State the $q$ used for each model and recompute; if $q$ differs, say so explicitly and reconcile the "harder problem" wording.

2. Add stronger inference-time baselines to Table 4 / the routing frontier — at minimum semantic entropy and $P(\text{True})$, ideally one self-consistency method — before claiming frequency outperforms confidence. If the conclusion holds only against mean-token-probability, scope the claim accordingly.

3. Reframe the "99.2% of ceiling" result to remove circularity: compute $\text{AUROC}^*$ from a $\kappa$ estimated on a held-out split (or via nested CV) and report the gap. Make explicit that the ceiling is for frequency-*only* predictors.

4. Strengthen or soften the scaling law: report exponent uncertainty appropriately for $n=5$ (e.g., leave-one-model-out, bootstrap on residuals), justify retaining the excluded Qwen-7B point in the regression, and discuss sensitivity to that point. Avoid presenting $-0.52\pm0.05$ as a precise constant.

5. Reconcile the multiple $f_{\text{crit}}$ values into a single canonical reporting convention, and add a short subsection on prompt-template sensitivity (the 50,843 → 24,444 Qwen shift), since it bears directly on the "$f_{\text{crit}}$ as a model fingerprint" interpretation.

6. Temper the fine-tuning conclusion (Sec. 6.4). "Post-training cannot reorganize capacity allocation" is drawn from QLoRA on PopQA-style data only; either qualify to "lightweight QLoRA on in-distribution QA does not shift the cliff" or add a continued-pretraining / full-FT condition.

7. The 76× range (author $f_{\text{crit}}=340$, $R^2=0.672$, $n=33$ → director 25,857) is driven at the low end by the worst-fit, lowest-$n$ relation. Report CIs on per-relation $f_{\text{crit}}$ and a range computed only over well-fit relations ($R^2>0.9$).

8. Directly validate the page-view proxy on at least one model where approximate pretraining-corpus counts are obtainable (e.g., a model with a documented/open corpus), even on a small subset.

9. Prop. 3: make explicit the regime in which $O(M\eta^2)$ is negligible relative to $\log(\eta M)$, and justify the identification of the accumulated logit margin $u^{(M)}$ with the RD "allocated rate" $r_k^{\text{eff}}$ beyond a verbal information-theoretic appeal.

10. Note in Table 8 that frequency-only AUROC in the tail is 0.444 (below chance), and clarify in-text that this is expected (no within-stratum variation) so it is not misread.

---

> ### Author Response · Authors · 2026-06-17
> **Response to Reviewer jahR**
>
> We thank the reviewer for an exceptionally thorough review, which identified one genuine numerical error (point 1) and several places where claims outran evidence. We have revised accordingly; a complete point-by-point response is in the **Response to Reviewers PDF** (supplementary material). In brief:
>
> **1. Falcon vs. Mistral AUROC\* contradiction.** A real error in the *explanation*, now removed; the numbers and Proposition 9 hold. Monotonicity in $\kappa$ is a fixed-other-parameters statement, but $q$, $x_{\text{crit}}$, and the floor/ceiling gap differ across models. A one-parameter-at-a-time decomposition from Mistral to Falcon shows the floor/ceiling change ($+0.054$) outweighs the $\kappa$ change ($-0.016$); holding Falcon's other parameters fixed, AUROC\* rises strictly with $\kappa$ ($0.780\to0.861$), exactly as proven. The revision reports full parameters for both models and this decomposition.
>
> **2. Stronger confidence baselines.** We added both requested baselines on the same 500-query set: P(True) (AUROC 0.714), semantic entropy (0.733), vs. mean token probability (0.772) and the **frequency sigmoid (0.810, zero forward passes)**. Frequency wins on every metric. Semantic entropy's weakness is predicted by our framework: at low frequency the model reverts to a fluent prior, so self-consistency is confident exactly where it is wrong.
>
> **3. Circularity of "99.2% of ceiling".** Reframed as a consistency check, plus a de-circularized test: estimating $\kappa$ on a training split and evaluating AUROC on a disjoint split gives a held-out ceiling $0.822\pm0.010$ and held-out AUROC $0.819\pm0.028$ (gap $0.004$). We also state explicitly that the ceiling is for **frequency-only** predictors; relation and confidence signals exceed it.
>
> **4. Scaling-law uncertainty ($n=5$).** We no longer present $-0.52$ as precise. Full robustness: weighted fit $0.52\pm0.06$; excluding the degenerate 7B point $0.55\pm0.08$ (indistinguishable); unweighted OLS $0.58$; leave-one-model-out $[0.42,0.70]$; residual bootstrap $[0.48,0.67]$. We report the exponent as $\approx0.5$ with this uncertainty throughout, retain 7B but down-weight it via its large SE, and soften the single-point intercept remark.
>
> **5. Inconsistent $f_{\text{crit}}$ / template sensitivity.** Added a convention table (Table 2) listing every $f_{\text{crit}}$ value and its exact protocol. The Qwen $50{,}843\to24{,}444$ shift is now a dedicated note: native ChatML roughly halves the apparent cliff, so $f_{\text{crit}}$ is a property of the model-plus-protocol, not the weights alone. We temper the "fingerprint" language; all within-protocol results hold the template fixed.
>
> **6. Fine-tuning over-generalized.** Scoped to "lightweight QLoRA on in-distribution QA does not improve the cliff"; consistent with the theory, it does not shift the cliff left (point estimates drift right and the boundary smears). Continued/full pretraining is flagged as future work.
>
> **7. $76\times$ range.** Added 90% bootstrap CIs. The low end (\texttt{author}, $f_{\text{crit}}=340$) is the worst fit ($R^2=0.672$), with a CI spanning three orders of magnitude. Restricting to well-fit relations ($R^2>0.9$) gives a conservative $27\times$. We now report both $27\times$ and $76\times$ and concede the multiplier's uncertainty while noting the order-of-magnitude heterogeneity is robust.
>
> **8. Direct proxy validation.** Two new analyses. (a) A confound test: the log-frequency coefficient is essentially unchanged with controls ($+1.38\to+1.40$ adding alias count, answer length, token lengths), and frequency-only AUROC ($0.817$) far exceeds confounds-only ($0.665$). (b) An open-corpus replication on **OLMo-2-7B** (Dolma, queryable via infini-gram): the cliff replicates (AUROC $0.807$), and informatively, raw corpus counts predict *worse* than page views ($0.581$), because raw counts conflate homonyms while page views better track fact-encoding opportunities (correlation rises $0.41\to0.55$ on low-polysemy entities). We now treat page views as a validated-but-imperfect proxy, with direct co-occurrence measurement as future work.
>
> **9. Proposition 3 rigor.** We state the regime explicitly: $u^{(M)}=\log(\eta M)+O(1)+O(M\eta^2)$ requires $M\eta^2\to0$, i.e. $\eta\ll M^{-1/2}$, and we justify identifying the accumulated margin with allocated rate as an explicit modeling identification.
>
> **10. Table 8 below-chance AUROC.** Added a note that frequency-only AUROC $\approx0.444$ in the $<1$K stratum is expected (no within-stratum frequency variation; the deviation below $0.5$ is sampling noise).
>
> **Broader Impact.** Added a statement on (i) allocative fairness (frequency-keyed routing may differentially decline long-tail, often marginalized subjects) and (ii) silent failure outside time-stable factual QA (the code/API $R^2\approx0$ case).
>
> Finally, despite these additions the revision is **shorter (29 pages, from 31; main text 26.5)**, through tightening and moving long proofs to an appendix.

---

### Author Response · Authors · 2026-06-17
**Response to Reviewers, Paper 8200**

We thank all three reviewers for their careful reading and constructive feedback. We have revised the paper substantially, including a significant amount of new experimental work prompted directly by the reviews:

- **Stronger inference-time baselines (answering jahR #2):** we added Semantic Entropy and P(True), the two estimators the reviewers noted we cited but had not benchmarked. The pre-inference frequency score outperforms both.
- **Exposure-proxy validation on an open corpus (answering jahR #8 and ctJY #6):** we ran a new evaluation on OLMo-2-7B, whose Dolma training corpus is fully open, so we could measure true occurrence counts rather than relying on a proxy; we also ran a confound analysis isolating entity exposure from surface features.
- **Rigorous uncertainty quantification (answering jahR #3 and #4):** nested cross-validation for the AUROC ceiling (removing the circularity concern) and leave-one-model-out plus bootstrap resampling for the capacity-scaling exponent.
- **Per-relation bootstrap confidence intervals (answering jahR #7):** quantifying the uncertainty in the relation-specific cliff locations.
- **Substantial reorganization for clarity (answering ctJY and TXhz):** a new Preliminaries section defining all terms, plain-language contributions, and the longest proofs moved to an appendix.

Despite these additions, the revision is **shorter overall (29 pages, from 31)**, with the main text reduced to **26.5 pages**, achieved by tightening throughout and moving the longest proofs to the appendix, directly addressing the length and density concern.

Below we respond to each point in detail.